# Sea-ice derived meltwater stratification slows the biological carbon pump: results from continuous observations

Wilken-Jon von Appen [1✉], Anya M. Waite[1,2], Melanie Bergmann [1], Christina Bienhold[1,3], Olaf Boebel[1], Astrid Bracher [1,4], Boris Cisewski [5], Jonas Hagemann [1], Mario Hoppema [1], Morten H. Iversen [1,6], Christian Konrad[1,6], Thomas Krumpen[1], Normen Lochthofen [1], Katja Metfies [1], Barbara Niehoff[1], Eva-Maria Nöthig [1], Autun Purser [1], Ian Salter [1,7], Matthias Schaber[5], Daniel Scholz [1], Thomas Soltwedel[1], Sinhue Torres-Valdes [1], Claudia Wekerle[1], Frank Wenzhöfer [1,3], Matthias Wietz [1,3] & Antje Boetius [1,3,6]

The ocean moderates the world's climate through absorption of heat and carbon, but how much carbon the ocean will continue to absorb remains unknown. The North Atlantic Ocean west (Baffin Bay/Labrador Sea) and east (Fram Strait/Greenland Sea) of Greenland features the most intense absorption of anthropogenic carbon globally; the biological carbon pump (BCP) contributes substantially. As Arctic sea-ice melts, the BCP changes, impacting global climate and other critical ocean attributes (e.g. biodiversity). Full understanding requires year-round observations across a range of ice conditions. Here we present such observations: autonomously collected Eulerian continuous 24-month time-series in Fram Strait. We show that, compared to ice-unaffected conditions, sea-ice derived meltwater stratification slows the BCP by 4 months, a shift from an export to a retention system, with measurable impacts on benthic communities. This has implications for ecosystem dynamics in the future warmer Arctic where the seasonal ice zone is expected to expand.

---

[1] Alfred Wegener Institute, Helmholtz Centre for Polar and Marine Research, Bremerhaven, Germany. [2] Department of Oceanography and the Ocean Frontier Institute, Dalhousie University, Halifax, NS, Canada. [3] Max Planck Institute for Marine Microbiology, Bremen, Germany. [4] Institute of Environmental Physics, University of Bremen, Bremen, Germany. [5] Thünen Institute of Sea Fisheries, Bremerhaven, Germany. [6] MARUM, University of Bremen, Bremen, Germany. [7] Faroe Marine Research Institute, Tórshavn, Faroe Islands. ✉email: Wilken-Jon.von.Appen@awi.de

Phytoplankton require light to flourish. Stratification keeps algal cells in surface layers. In the Arctic Ocean, this is often controlled by sea ice melt and brine release. The extent of stratification, often simplified as mixed layer depth (MLD), impacts the timing of biological production in nutrient-rich open waters[1], at the ice edge on Arctic shelves[2], and under the ice[3]. For example, in a MLD of 20 m, a bloom can start 4–6 weeks earlier than in a MLD of 100 m[4]. Large interannual differences in the timing of blooms may be challenging for higher trophic levels due to temporal and spatial mismatches with key prey, and could therefore impact trophic interactions considerably[5]. Also, different pelagic bloom scenarios can result in different proportions of sinking algal aggregates, fecal pellets, and marine snow, and the sea-ice cover plays an important role in such scenarios in the Arctic[6,7]. Consequently, the structure, sinking rate, biogeochemical composition, and nutritional quality of particles reaching the benthic community may change with Arctic warming and sea-ice retreat[8].

Stratification affects both light availability and nutrient supply as essential factors for phytoplankton growth. Typically, the open ocean regions of the Arctic are characterized by a well-stratified surface layer with rather low nutrient concentrations, and limited replenishment, leading to fast depletion upon the onset of the productive season[9]. Subsurface waters, however, are richer in nutrients[9,10] and upwelling of these nutrients at the ice edge may support patchy phytoplankton blooms, which can account for half of the regional production within a season[11]. When the ice recedes during spring/summer, ice-edge blooms may form in the warm water in the Nordic and Barents Seas[12] and move with the receding ice. In the Greenland Sea, three successive phases of phytoplankton growth were identified: first under ice, then at the ice edge, and finally in the open water with a subsurface chlorophyll *a* maximum[3]. North of the Greenland Sea in Fram Strait, primary production can change by 0.3 g C m$^{-2}$ day$^{-1}$ between years, with higher values occurring in years when more sea ice is exported southwards from the Arctic Ocean[13,14].

The understanding of the ocean's role in the global carbon balance and in response to climate change is an urgent scientific task[15]. The North Atlantic-Arctic gateways are sites of intense climate-induced changes affecting food-webs and carbon export[16,17]. Thus, moving from the concept of a single spring bloom to a nuanced and mechanistic understanding of Arctic productivity including the role of sea ice is essential for our understanding of the biological carbon pump (BCP) at high latitudes.

The FRAM (Frontiers in Arctic Marine Monitoring) Ocean Observing System in Fram Strait[18] provides continuous in-situ data that can help bridge key gaps in ecosystem research with regard to temporal dynamics. Two mooring clusters, equipped throughout the water column with a comprehensive suite of physical and biogeochemical sensors as well as autonomous sampling systems, are located in central (mooring cluster "HG-IV") and eastern (mooring cluster "F4") Fram Strait (Fig. 1), the only deep-water gateway connecting the Arctic Ocean to the rest of the world's oceans. Relatively warm and salty Atlantic Water (AW) flows northward with the West Spitsbergen Current (WSC, Fig. 1, mooring location F4)[19] in eastern Fram Strait. Some of this AW flows westward in an energetic eddying recirculation (mooring location HG-IV)[20] and subducts below the colder and fresher Polar Water (PW) exiting the Central Arctic Ocean. This subduction stirs and mixes AW and PW on small horizontal scales[21]. The southward flowing PW carries sea ice from the Transpolar Drift and forms the East Greenland Current (EGC, Fig. 1). The confluence of the AW (which can melt sea ice) and the southward flowing sea ice results in a semi-stationary ice edge

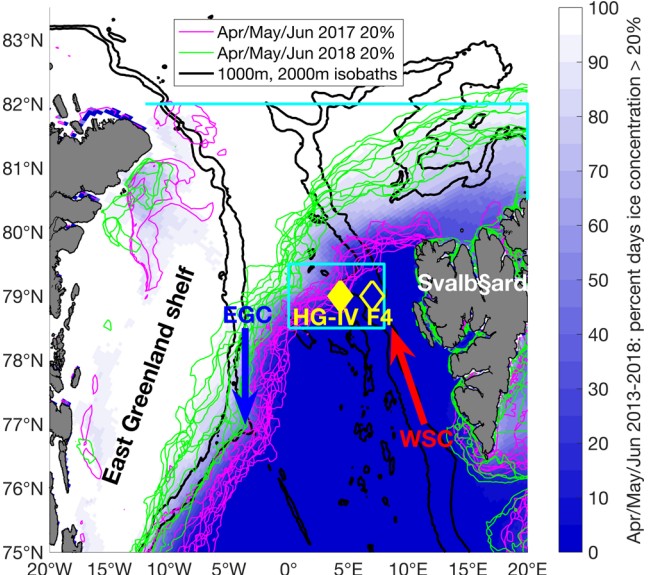

**Fig. 1 Map of mooring locations, major currents, and ice coverage in Fram Strait.** The percentage of days in April/May/June 2013–2018 during which sea ice cover exceeded 20% is shown in white-blue. The width of the marginal ice zone (20% contour to 80% contour) was typically less than 50 km. The weekly 20% sea ice concentration contours for the study period in April/May/June are shown in magenta (2017) and green (2018). The meltwater regime typically applies in the area covered by the variability of the 20% contours. The mixed layer regime by contrast applies well east/south (~50–100 km) of the 20% contours. Thus mooring HG-IV is in the meltwater regime in 2017 and in the mixed layer regime in 2018. The major currents in the area are indicated schematically: West Spitsbergen Current (WSC) and East Greenland Current (EGC). The location of the moorings discussed in this study are marked in yellow: F4 in the WSC (data shown in Figs. S1–S3) and HG-IV west of the WSC (data shown below). The 1000-m and 2000-m isobaths are shown in black and land in gray. The gate and box used in Figs. 2 and 9, respectively, are shown in cyan.

(here defined as 20% sea ice concentration, but not sensitive to the exact definition). Compared to other regions of the Polar oceans, the seasonal migration of the ice edge of 50–100 km in Fram Strait is small[22], meaning the seasonal ice zone is narrow. It covers an area of approximately 40,000 km$^2$ and, due to its high productivity, forms an ecologically important feeding ground for marine birds and mammals[23].

Here we investigate the effect of varying sea ice export from the Arctic Ocean on local biogeochemical processes and the biological carbon pump. We hypothesize that high ice-export results in strong meltwater stratification leading to the formation of a thin productive layer near the surface. By contrast, low sea ice export results in a classical (deeper) mixed layer which features a bloom of pelagic diatoms that is rapidly exported to the seafloor. Building on work that was partially able to achieve this, we tested the hypothesis that sea ice export is a key regulating factor not only for the biological carbon pump but also for the plankton community composition and retention efficiency.

## Results

**Ice export and related stratification regimes.** Key to understanding processes at our mooring sites is their location in a given year relative to the ice extent in the marginal ice zone of Fram Strait (Fig. 1). Ice extent is driven by southward sea ice export—the Transpolar Drift—from the Arctic Ocean proper across Fram Strait, as well as by local winds and currents. This sea ice export

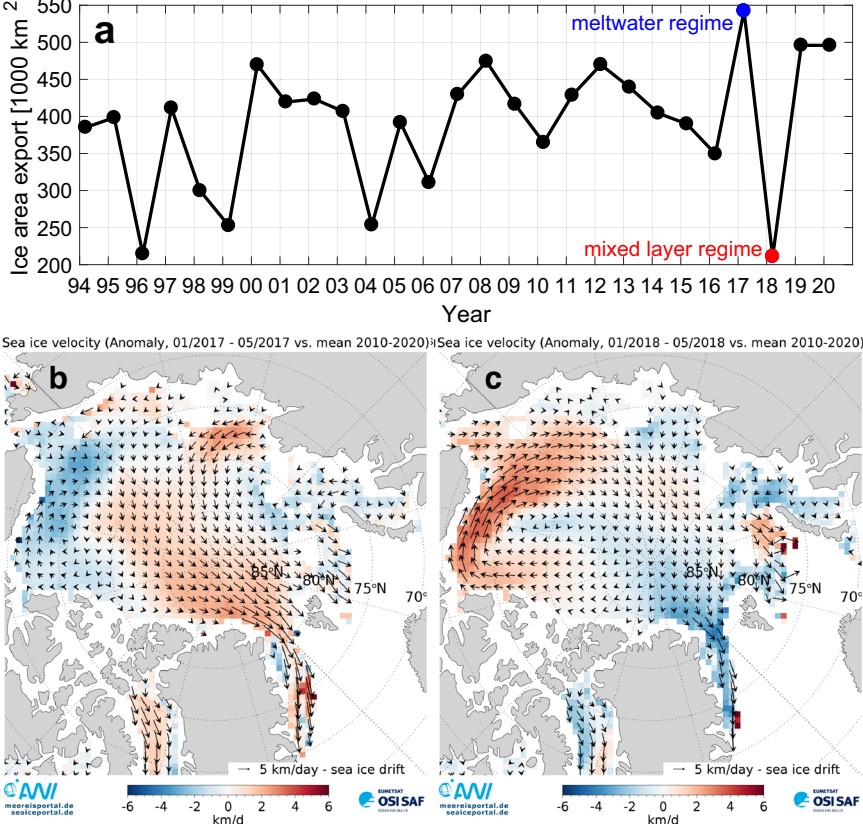

**Fig. 2 Fram Strait winter ice export and Arctic winter sea ice velocity anomalies. a** Southward sea ice area export [1000 km$^2$] in winter (January to May) between 1994 and 2020 across a zonal gate at 82°N between 12°W and 20°E and a meridional gate at 20°E between 80.5°N and 82°N (see cyan gate in Fig. 1) as used in ref. [105]. The winter values preceding the two discussed stratification regimes are marked. **b**, **c** Sea ice velocity [km day$^{-1}$] vectors and anomaly from 2010–2020 mean in color. **b** is January to May 2017 average and **c** is January to May 2018 average. Note that the sea ice velocity north of Fram Strait was anomalously large in 2017 and anomalously small in 2018.

in turn is driven by the atmospheric surface pressure gradient between Greenland and Svalbard[24]. As a result, there are substantial interannual differences in ice export through Fram Strait. Overall, Arctic warming has increased Transpolar Drift velocity[25], which in turn has led to an intensified sea ice area export out of Fram Strait in winter (January–May, Fig. 2a). However, an exact quantification of trends in export rates remains difficult, as satellite-based motion estimates show high uncertainties at export gateways[26]. The ice area export in 2017 was anomalously large while it was anomalously small in 2018 (Fig. 2b, c). Consequently, in spring/summer 2018, the ice edge was >50 km to the northeast of mooring HG-IV in the central Fram Strait while the ice edge was above the mooring in spring/ summer 2017 (Fig. 1, see below).

The presence of ice in 2017 provided a source of meltwater that resulted in a very strong ($N^2 = 1 \times 10^{-4}$ s$^{-2}$), but temporally variable haline (salinity driven) stratification at mooring HG-IV as estimated from our time series observations between 30 m (the shallowest depth that was deemed safe for moored observations) and 55 m (Figs. 3a, c and 4e). However, shipboard observations showed that stratification between 0 m and 30 m in fact was 10-fold stronger than in the layer between 30 m and 55 m (Fig. 3a, c). Surface salinities as low as 30.5 were observed in the top 3 m in July 2017 when meltwater contributed up to 1/6 by volume assuming that the surface water originated only from sea ice melted in Atlantic Water. Furthermore, there was no mixed layer (or if it was there, it must have been <5 m thick), similar to what has been reported as summer conditions across the central Arctic Ocean[27].

By contrast, there was a much weaker stratification across a mixed layer of approximately 50 m at the same location and time period in spring/summer 2018 (Figs. 3b, d and 4c). Most of that stratification was also due to the vertical salinity gradient (Fig. 4e). Winter-time deep mixing occurred in both years and restored nutrient concentrations in the surface (Fig. 5e), but in 2017 it was intermittently interrupted by meltwater advection as early as March into the region. This ceased on ~15-May-2017 as the surface heat flux turned positive (Fig. 4b), i.e., the atmosphere started to warm the ocean. In the following year, deep mixing persisted from ~15-Dec-2017 until 01-May-2018 when the surface heat flux turned positive (Fig. 4b), and a period of relatively weaker, but uninterrupted stratification started. 2018 had the highest May air temperatures in Svalbard since 1898[28].

Our time series thus covered two distinct stratification regimes during the spring blooms in the two-year observation period (summer 2016 to summer 2018): Strong salinity stratification due to meltwater (spring/summer 2017, "MW regime") and relatively weak stratification with a significant mixed layer (spring/summer 2018, "ML regime").

The mooring measurements in our high-resolution time series must be understood within this changing oceanographic context, particularly in the vertical dimension. As we demonstrate below (and summarize in Table 1), the strong meltwater-induced stratification in 2017 resulted in primary production at depths above the moored sensors and samplers located at 30 m depth. By contrast, in 2018, primary production took place in a weakly stratified mixed layer and the upper measurement depth (30 m) was within this productive mixed layer.

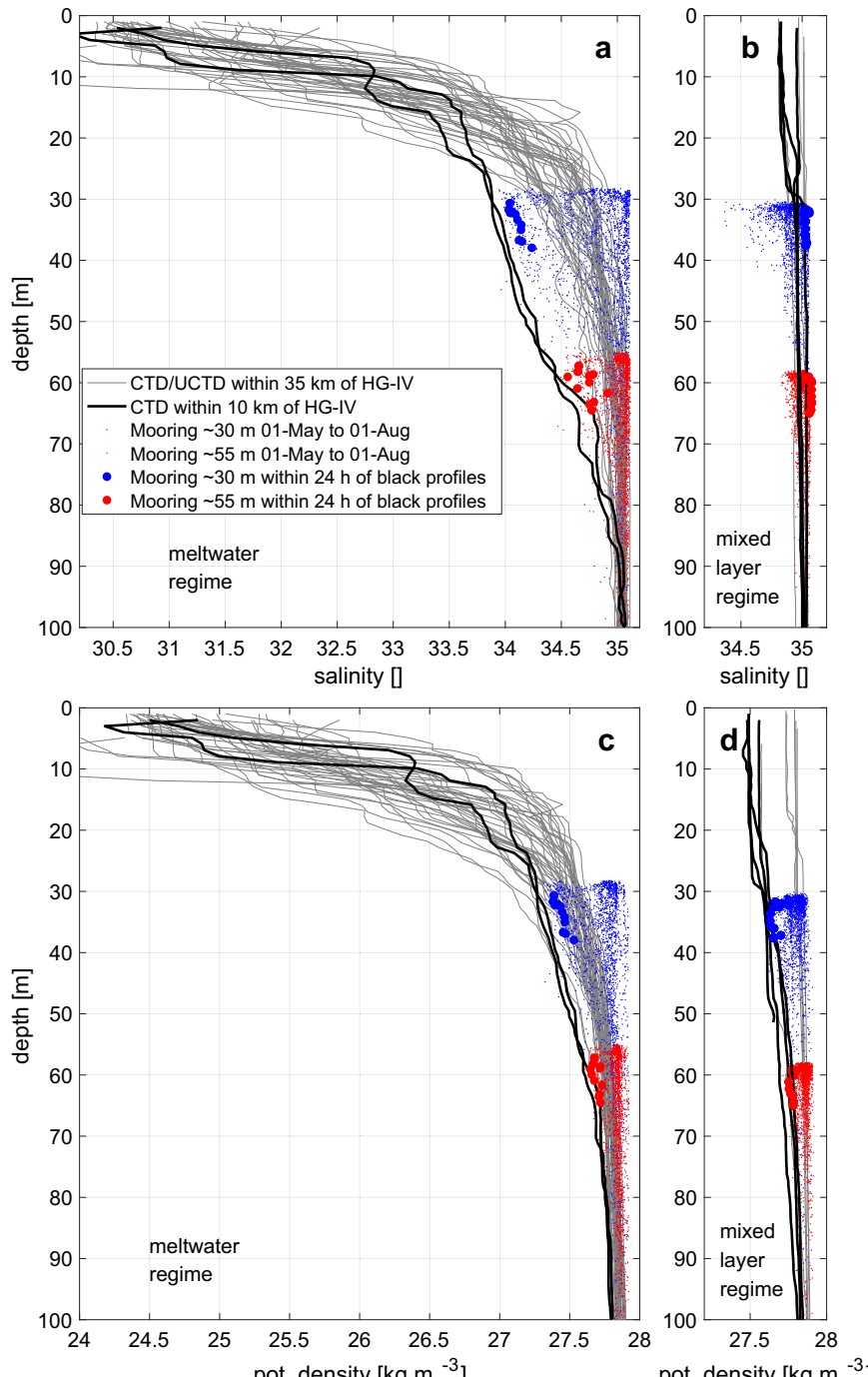

**Fig. 3 Upper ocean stratification from comparison of shipboard and mooring data. a**, **b** Salinity [] and **c**, **d** potential density [kg m⁻³] in 2017 (meltwater regime in **a**, **c**) and 2018 (mixed layer regime in **b**, **d**). All CTD/UCTD casts within 35 km of HG-IV mooring location (gray lines). All CTD casts within 10 km of HG-IV (black lines). All mooring measurements between 01-May and 01-Aug at ~30/~55 m (blue/red dots). Mooring measurements within 24 h of black profiles (large blue/red dots). Note that the aspect ratio of the subplots of the respective parameters is identical.

**Biogeochemical and biological characteristics of the blooms and the associated carbon export.** We use the detailed biogeochemical time series provided by our mooring data (Tables S1–S3) to approximately quantify total primary productivity and estimate seasonal new production (nitrate driven primary production) for the two phytoplankton blooms associated with the contrasting sea ice and stratification regimes detailed above.

*Mixed layer regime (spring/summer 2018).* The ML regime that we observed during spring/summer 2018 is typical for conditions when the upper ocean is mostly unaffected by sea ice (the ice edge was >50 km from our observation location during this period, Fig. 4a). The high winter nutrient values (Fig. 5e) indicate that they have been resupplied by deep mixing down to more than 250 m, unlike the situation in western Fram Strait where the constant halocline is associated with a deep and permanent nitracline[29]. From the end of the Polar night in mid February until the end of April, the average light level experienced by a phytoplankton cell in the >250 m deep mixed layer (Fig. 4c) was too low to support growth (<10 µmol m⁻² s⁻¹, Fig. 5a red). As

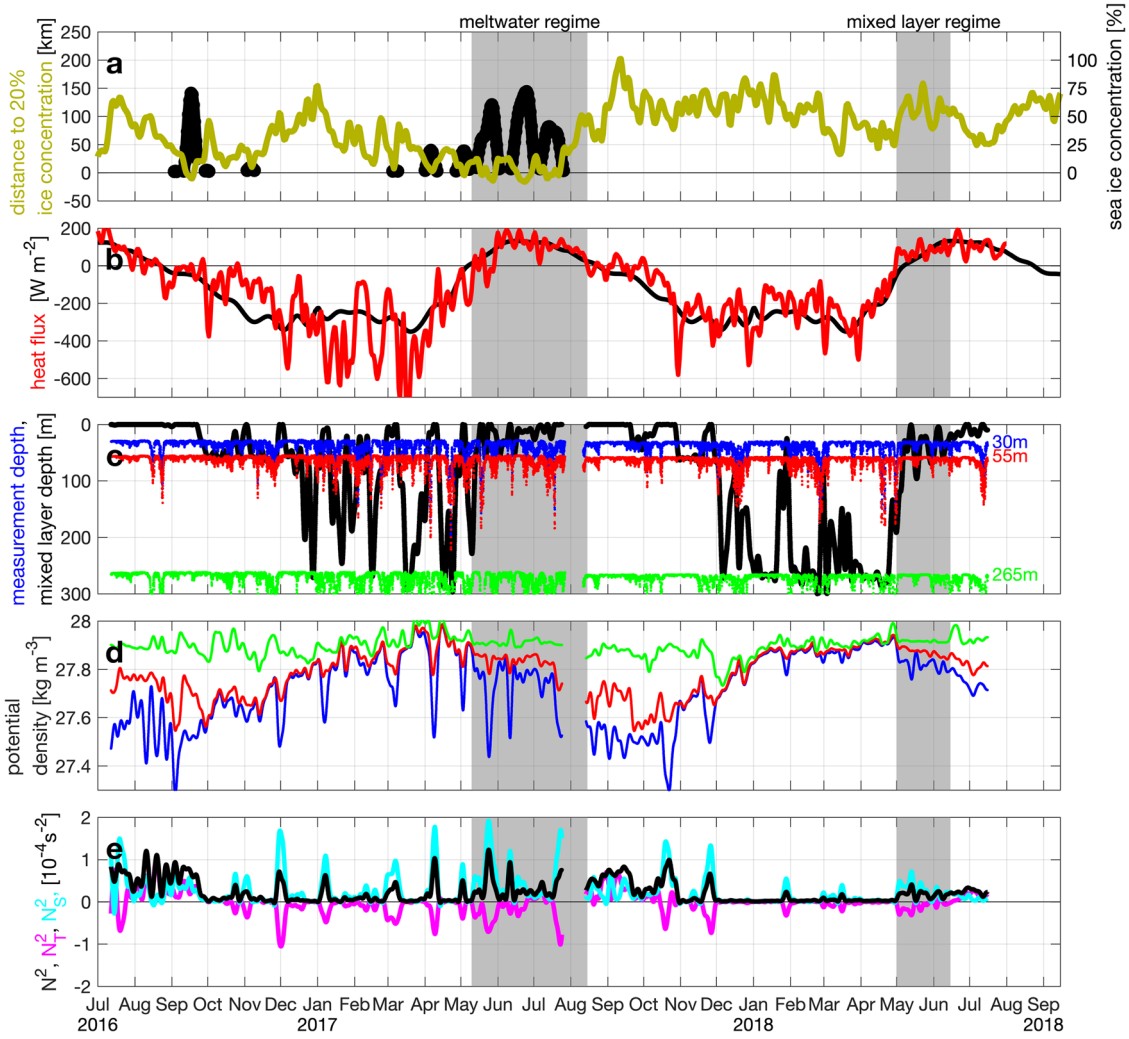

**Fig. 4 Ice, heat flux, and upper ocean stratification at HG-IV. a** Distance of mooring to 20% sea ice concentration [km] (dark yellow), negative when sea ice concentration at mooring exceeded 20%; sea ice concentration [%] (plotted black when >0%) at AMSR-2 satellite grid point (~5 km diameter) closest to mooring location; note the different $y$-scale for the sea ice concentration. **b** Net surface heat flux [W m$^{-2}$] (red) at ERI-I reanalysis grid point (~30 km diameter) closest to mooring location and its smoothed climatology (black); positive values warm the ocean. **c** Measurement depths of sensors [m] at approximately 30 m (blue), 55 m (red), and 265 m (green); minimum estimate of the mixed layer depth [m] (black), actual mixed layer depth was likely somewhat deeper. **d** Potential density [kg m$^{-3}$] at the three measurement depths as colored in Fig. 4c. **e** Stratification (buoyancy frequency $N^2$ [s$^{-2}$], black) estimated from difference of 30 m and 55 m observations and temperature ($N^2_T$, magenta) and salinity ($N^2_S$, cyan) contributions to stratification; note that stratification above ~30 m in 2017 was likely 10-times larger than values given here (Fig. 3). The bloom periods of the meltwater (10-May to 15-Aug-2017) and mixed layer (01-May to 15-Jun-2018) regimes are marked by gray backgrounds. All sensor data shown in Figs. 4–6 are hourly data that have been lowpass filtered with a 5-days cutoff.

the moderate stratification started around 01-May-2018, the photosynthetically available radiation (PAR) in the mixed layer increased by about an order of magnitude (Fig. 5a). This induced a bloom of pelagic diatoms—evident from both sequence abundances (Fig. 6a) and microscopic cell counts[30] and an associated decrease of nitrate and pCO$_2$, an increase in chlorophyll and pH (Fig. 5b–e), and an increase in photosynthesis-derived oxygen, leading to a decrease in apparent oxygen utilization (AOU). The early bloom approximately followed Redfield stoichiometry with ~7 μmol l$^{-1}$ of oxygen produced, ~6 μmol l$^{-1}$ of carbon taken up, and ~0.6 μg l$^{-1}$ of chlorophyll $a$ produced per μmol l$^{-1}$ nitrate taken up (Fig. 7). Note that oxygen will leave the mixed layer to the atmosphere while carbon dioxide will enter the mixed layer and therefore our estimates of 7 μmol l$^{-1}$ of oxygen produced and 6 μmol l$^{-1}$ of carbon taken up per μmol l$^{-1}$ nitrate represent lower bounds.

Chlorophyll $a$ (Fig. 5b) in the mixed layer increased with e-folding growth rates of ~0.2 day$^{-1}$ (Fig. S4d) eventually reaching concentrations of 7 μg l$^{-1}$. These values appear consistent compared to most previous observations in the region (mean of 4.8 μg l$^{-1}$ for 1991–2015 with a maximum of 7.2 μg l$^{-1}$ measured in May 1997)[31]. The chlorophyll increase coincided with shading by the increased turbidity from growing phytoplankton reducing PAR levels at 30 m to values equivalent to those at the end of the preceding Polar night (Fig. 5a) consistent with the expected euphotic depth at those chlorophyll concentrations of ~15 m[32,33]. The diatom bloom coincided with elevated relative abundances of *Bacteroidetes* (Fig. 6b) that likely utilized algal substrates, resembling bloom dynamics and metabolic interrelations in temperate regions[34]. The slightly lower oxygen production at 55 m (compared to 30 m) suggests this depth is at or below the base of the productive mixed layer. We therefore

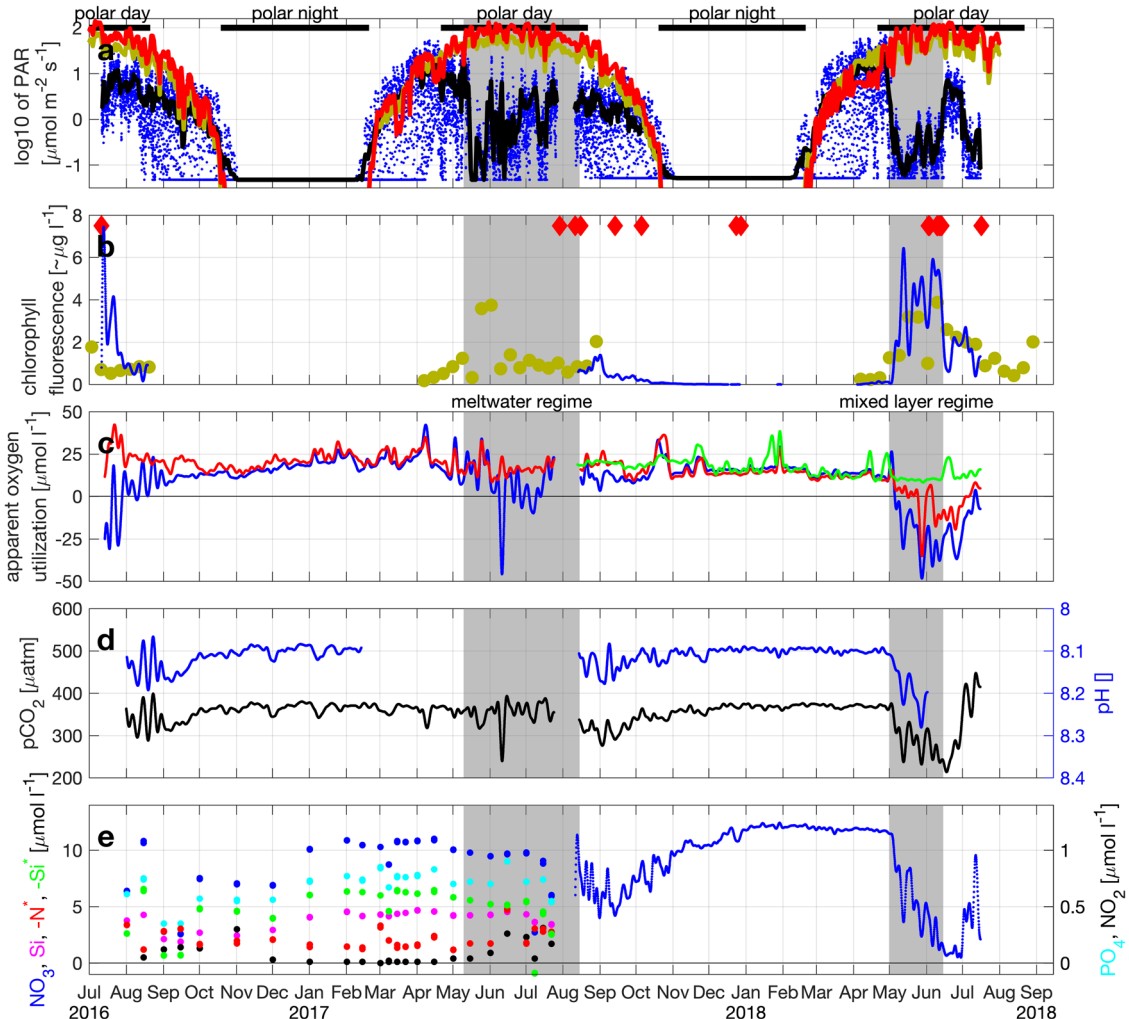

**Fig. 5 Upper ocean biogeochemistry at HG-IV from upper instrument depth (~30 m). a** Logarithm of photosynthetically available radiation [µmol m$^{-2}$ s$^{-1}$] from hourly sensor measurements at depth (blue) and their 5-day average (black); from reanalysis at the surface reduced to 30 m depth using a constant extinction coefficient in order to make it comparable to the sensor measurements (dark yellow); and from the reanalysis value distributed over the mixed layer depth (red). **b** Chlorophyll *a* concentration [~µg l$^{-1}$] from chlorophyll fluorescence sensor measurements (blue); and from average of Sentinel 3 A OLCI satellite grid points within 30-km radius of mooring location (dark yellow); times when individual 1-second measurements of scattering exceeded 0.002 m$^{-1}$ sr$^{-1}$ (red diamonds) indicative of aggregation. **c** Apparent oxygen utilization [µmol l$^{-1}$] at the three measurement depths as colored in Fig. 4c. **d** pCO$_2$ [µatm] (black) and pH [] (blue); note the reversed *y*-axis for pH. **e** Inorganic nutrients [µmol l$^{-1}$] from water samples in 2016–2017 and from nitrate sensor in 2017–2018; nitrate (blue), silicate (magenta), phosphate (cyan), nitrite (black); negative of (N*=nitrate-phosphate*16) (red); negative of (Si*=silicate-nitrate) (green); note the different *y*-scale for phosphate and nitrite.

estimate that the ML is on the order of 50 m. Integrating nitrate (~12 µmol l$^{-1}$) over this depth indicates the standing stock available at the onset of the productive season is 0.6 mol m$^{-2}$ of nitrate (Table 1). This is before considering likely resupply by mixing across the base of the mixed layer, and would—assuming the Redfield ratio—potentially support a total production of ~50 g m$^{-2}$ of carbon (Table 1). Three different estimates of bloom production from three synchronous data sets (pCO$_2$, nitrate, and maximum chlorophyll concentration combined with a chlorophyll to carbon ratio of 50; Table 1) result in estimates of ~72, 50, and 13 g m$^{-2}$ of carbon (Table 1). Note that the estimate based on chlorophyll converted to carbon is a lower bound. It neglects that some organic matter export will happen before the maximum in observed chlorophyll is reached and that further growth will occur after the maximum has been reached. We attribute this small range between the different estimates to the conversion factors used and assumptions inherent in the different methods, and consider it a strength of the high-resolution time

series, where the temporal variability is fully constrained and key events are not missed.

The bloom in the ML regime (2018) was dominated by pelagic diatoms (*Chaetoceros* and *Thalassiosira*) (Fig. 6a). Associated with the bloom onset, the zooplankton and fish biomass as estimated by acoustic backscatter quickly increased within a few days (Fig. 6c). It appears plausible that the suspension feeding herbivorous zooplankton responded directly to the diatoms which are their preferred food source[35,36] and that they produced fast-sinking fecal pellets exporting particulate organic carbon (POC) early in the season as measured in multiple depths (Fig. 6d). Chloroplasts can remain intact during consumption by zooplankton[37] consistent with the green color of the sedimented material observed on the seafloor (Fig. 6e). In addition, the diatoms likely formed fast-sinking aggregates (aggregation marked in red in Fig. 5b).

Within 2–3 weeks after the bloom onset, particles had reached 200 m (Fig. 6d) and 1200 m depth (not shown). A major POC

**Table 1 Physical and biogeochemical observations over the bloom duration.**

| Parameter | Unit | Meltwater regime (spring/ summer 2017) | Mixed layer regime (spring/ summer 2018) | Comments on 2017 | Comments on 2018 |
|---|---|---|---|---|---|
| Sea ice concentration | % | 0–75 | 0 | | |
| Distance to sea ice edge | km | 0–50 | >50 | | |
| Average mixed layer thickness | m | <10 | ≳50 | The meltwater resulted in a very strong salinity stratification, hence there was no real mixed layer | |
| Average productive layer thickness | m | ≪30 | ≳50 | Possibly in ~10 m thin layers | Inferred from less decline in AOU at 55 m |
| Bloom start | | 10-May | 01-May | | |
| Bloom end | | maybe ~15-Aug | 15-Jun | Observations do not show a classical bloom which crashes at a certain time | Bloom crashes as nitrate is used up |
| Bloom duration | months | ~3 | 1.5 | Observations do not show a classical bloom which crashes at a certain time | |
| Maximum chlorophyll $a$ concentration | gChl m$^{-2}$ | ≪0.12 | >0.3 | Based on upper range of chl satellite values and 30 m productive layer | Based on upper range of chl sensor values at 30 m and 50 m productive layer |
| Integrated apparent oxygen utilization (AOU) | μmol l$^{-1}$ | (~−15) | ~−60 | Observations at 30 m are given in () as they are below or at the very bottom of the productive layer and are therefore not representative of the productive layer. Vertically integrated values can therefore not be calculated. | In productive layer |
| Maximum pCO$_2$ drawdown | μatm | (~50) | ~180 | | |
| Integrated carbon takeup | μmol l$^{-1}$ molC m$^{-2}$ gC m$^{-2}$ | (~17) | ~124 6 72 | | In productive layer; Vertically integrated over 50 m productive layer |
| Integrated nitrate takeup | mol m$^{-2}$ | ≪0.3 | >0.6 | No drawdown seen at 30 m; upper bound estimated from nitrate concentration at 30 m | Lower bound due to likely resupply by mixing |
| Net primary production from nitrate takeup | molC m$^{-2}$ gC m$^{-2}$ | ≪2.1 ≪25 | >4.2 >50 | Upper bound as mixing inhibited. In addition: regenerated primary production from ammonia (not observed) | Lower bound due to likely resupply by mixing |
| Integrated export measured by lander trap over | gC m$^{-2}$ | 1.3 (Mar-Aug) 2.1 (Sep-Nov) | 3.4 (Mar-Aug) 1.1 (Sep-Nov) | | September 2018 data is not measured, but calculated as the average of September values in 2004–05, 07–11, 16–17. |
| Stoichiometry of temporal evolution of bloom | | | (7 μmolO$_2$ l$^{-1}$)/ (1 μmolNO$_3$ l$^{-1}$); (0.6 μgChla l$^{-1}$)/ (1 μmolNO$_3$ l$^{-1}$) | Time series observations are not in productive layer | From daily average values in productive layer 01May–15Jun |
| Phytoplankton carbon per chlorophyll | (μg l$^{-1}$)/ (μg l$^{-1}$) | | 6–60 | | Monotonically increases until bloom crashes[30] |

(gray background shading in Figs. 4–6) of parameters contrasting the meltwater and mixed layer regimes at HG-IV. Depending on the parameter, its average, maximum, range, or temporal integral is given.

export event of ~1 g m$^{-2}$ of carbon (~30 mg m$^{-2}$ day$^{-1}$ resolved for ~1 month; Fig. 6d) arrived at the seafloor in ~2600 m depth within 4–7 weeks after the onset of the bloom. Particle tracking in a numerical ocean model suggests the origin to be within 100 km horizontally[38] and therefore presumably from upper ocean conditions as described above. The export event covered up to 45% of the seafloor area with green detritus material (Fig. 6e) suggesting pigmentation, i.e., high chlorophyll $a$ levels, and thus likely rich in nitrogen and other nutrients. The sedimented material included pteropod shells indicative that some grazing likely had taken place. This was a significant deposition of material on the seafloor, which appeared to be characterized by high nutritional quality that attracted benthic megafauna and fish in considerable numbers (5-fold rise; Fig. 6e) and caused a 50% increase in benthic O$_2$ consumption (Fig. 6d).

By mid June, i.e., within 1.5 months after the start of the bloom, nitrate in the productive mixed layer was depleted (Fig. 5e) with decreasing chlorophyll $a$ concentrations in the euphotic zone (Fig. 5b) and increased light reaching 30 m depth (Fig. 5a). The system then transitioned to heterotrophic dinoflagellate dominance (data not shown), and nitrate, AOU, and pCO$_2$ all increased, likely through remineralization and

respiration, with *Bacteroidetes* continuing to account for a large proportion of the bacterial community (Fig. 6b).

*Meltwater regime (spring/summer 2017).* In 2017, ice export from the Arctic Ocean was enhanced and correspondingly ice cover and subsequent ice melt in Fram Strait were large. Thus the bloom phenology in the meltwater-stratified water column (MW regime) of the year 2017 was fundamentally different from the bloom phenology during the ML regime in 2018. The reduction of PAR at 30 m depth started later (mid May) and lasted only for about half a month (Fig. 5a), but was similar in amplitude to 2018. A reduction of PAR by one order of magnitude could be due to shadowing from ice if the open water fraction decreased by at least an order of magnitude, e.g., by a change from 0% to >90% ice concentration. However, ice concentrations did not exceed 75% during the MW regime (Fig. 4a), and PAR at 30 m was reduced by almost 2 orders of magnitude (Fig. 5a). Hence, we conclude that the shadowing resulted from a phytoplankton bloom that took place in the water column above the sensor depth (30 m). This is corroborated by satellite observations of increased surface chlorophyll $a$ (Fig. 5b) and increased nitrite concentrations at 30 m during that period (Fig. 5e).

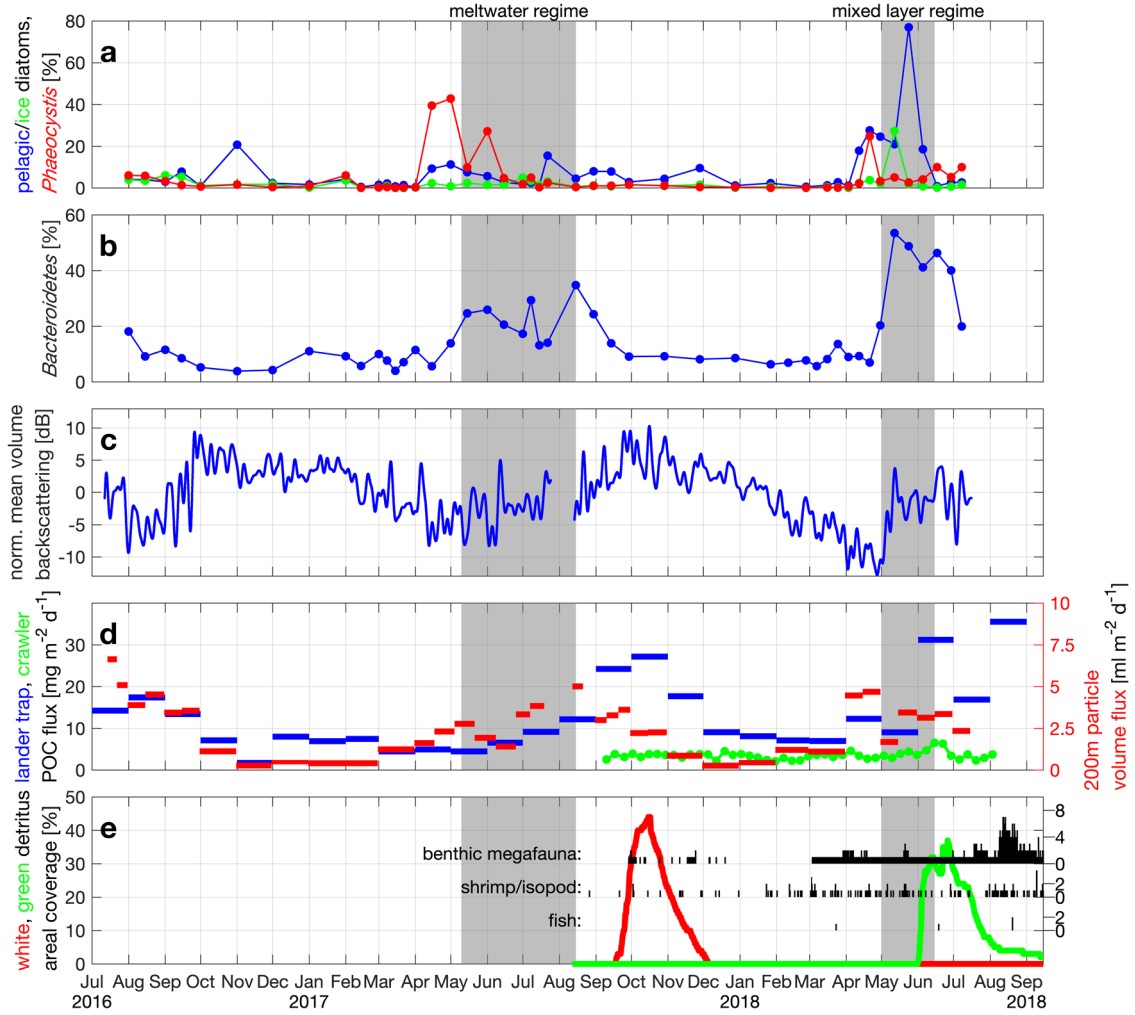

**Fig. 6 Upper ocean biology, export, and benthic response at HG-IV. a** Relative sequence abundances [% eukaryotes] of pelagic (*Thalassiosira* and *Chaetoceros*; blue) and ice-associated (*Fragilariopsis*; green) diatoms and *Phaeocystis* (red) at ~30 m. **b** Relative sequence abundance [% bacteria] of *Bacteroidetes* at ~30 m. **c** Normalized mean volume acoustic backscattering [dB] from 75 kHz ADCP (blue) averaged over 50–100 m depth range as a proxy for zooplankton biomass; see Fig. S5 for a comparison to shallower reaching multi-frequency AZFPs. **d** Particulate organic carbon flux [mg m⁻² d⁻¹] from lander sediment trap at 2 m altitude (blue) and from benthic crawler Tramper's oxygen microprobe measurements at the seafloor (green); particle volume flux [ml m⁻² d⁻¹] from traps in 200 m depth (red). **e** Percentage of seafloor covered by fine white detritus (red) and coarser green detritus (green); number of epibenthic megafaunal organisms, shrimp/isopod, and fish in photos indicated as vertical black lines with separate *y*-scales on the right.

Considering the extremely strong stratification to within less than 5 m of the surface encountered in the shipboard observations (Fig. 3a, c), we infer that this early primary production resulting in substantial turbidity took place in a thin layer (possibly as thin as 10 m) that limited light in the water below. A euphotic depth of 20 m is achieved by an average chlorophyll concentration of 4 µg l⁻¹ (as measured by the satellite, Fig. 5b) throughout the euphotic layer[32]. Considering that the satellite possibly underestimates the concentration below the surface and that it averages somewhat in space and time, 20 m is an upper bound on the euphotic depth supporting the speculation that the productive layer may have been as thin as 10 m. We assume that when the nitrate was used up in the surface, the productive layer progressively moved downward but did not reach 30 m. The biogeochemical measurements at 30 m only show a very small reduction in nitrate and silicate and little production of oxygen, highlighting the very shallow nature of the bloom in highly stratified surface waters and not reaching 30 m. There is no reason to suspect that the nitrate concentration above 30 m was higher than at 30 m. Hence, we take the 30-m nitrate concentration and integrate it over the productive layer thickness

to arrive at an upper bound on the column-integrated nitrate available for production in the highly stratified top layer; it could have sustained <25 g m⁻² of carbon (Table 1), i.e., less than half of what was available in the ML regime (2018). Furthermore, resupply by vertical mixing was very unlikely due to the strong stratification (Fig. S6). Note that the oxygen peak and pCO₂ deficit of 10–12 of June 2017 (Fig. 5c, d) appear to be related to an advective event typical in the dynamic marginal ice zone[21].

The water samples indicate that the bloom in the MW regime (2017) was dominated by *Phaeocystis* spp. with contributions of pelagic and ice-associated (*Fragilariopsis*) diatoms (Fig. 6a). *Phaeocystis* spp. can be positively buoyant[39,40] supporting its presence throughout the very top of the water column. Furthermore, it is known to form shallow blooms in the marginal ice zone[41–43] and when not aggregated, its sinking speed is small (0.1–2 m day⁻¹)[39]. Since the water sampler was moored below the productive layer, we cannot quantify relative species abundance in the productive layer (Fig. 6a). *Calanus finmarchicus* dominated the biomass of the herbivorous zooplankton in the eastern Fram Strait[44] but scarcely grazes on single cell *Phaeocystis*[45]. Consistently, a response in the zooplankton

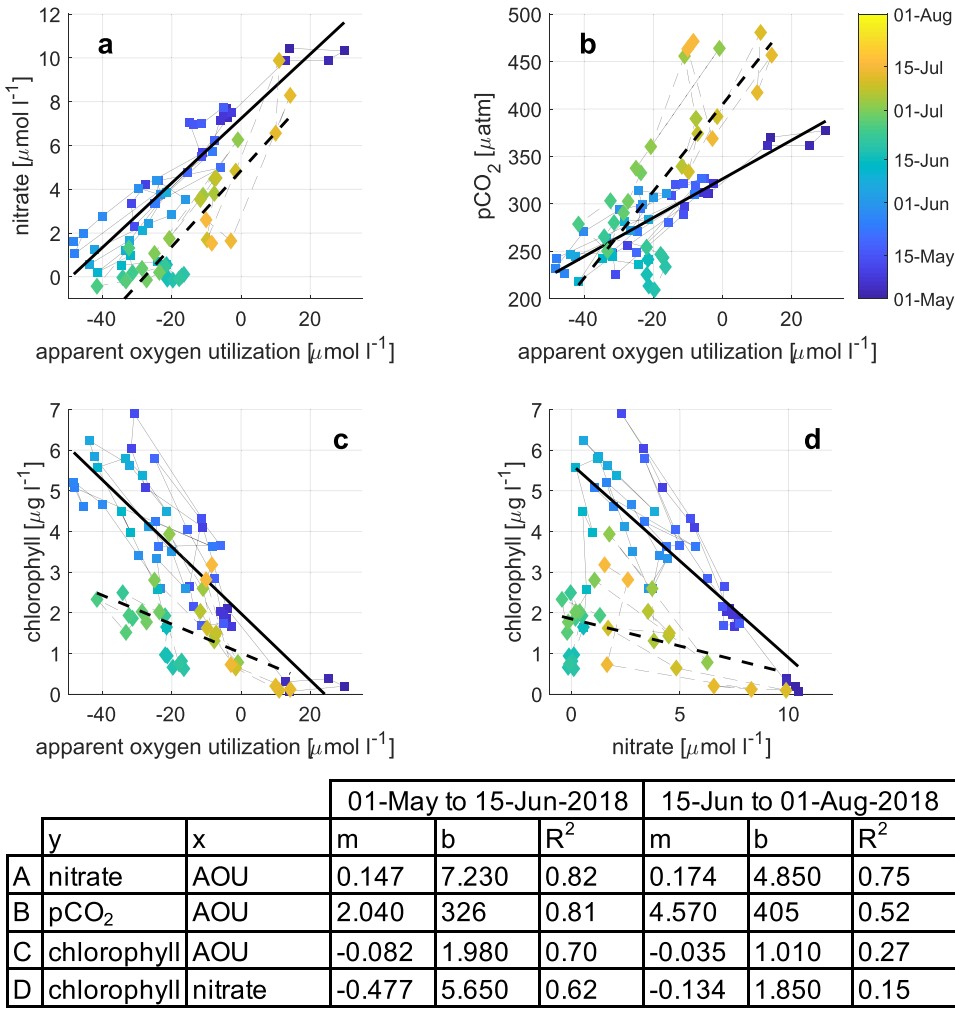

| | y | x | 01-May to 15-Jun-2018 | | | 15-Jun to 01-Aug-2018 | | |
|---|---|---|---|---|---|---|---|---|
| | | | m | b | $R^2$ | m | b | $R^2$ |
| A | nitrate | AOU | 0.147 | 7.230 | 0.82 | 0.174 | 4.850 | 0.75 |
| B | $pCO_2$ | AOU | 2.040 | 326 | 0.81 | 4.570 | 405 | 0.52 |
| C | chlorophyll | AOU | -0.082 | 1.980 | 0.70 | -0.035 | 1.010 | 0.27 |
| D | chlorophyll | nitrate | -0.477 | 5.650 | 0.62 | -0.134 | 1.850 | 0.15 |

**Fig. 7 Stoichiometry of bloom in mixed layer regime.** Scatter plots of daily averages of apparent oxygen utilization (AOU) [µmol l⁻¹], nitrate [µmol l⁻¹], $pCO_2$ [µatm], and chlorophyll [µg l⁻¹] colored by date in 2018 at HG-IV. **a** Nitrate vs. AOU, **b** $pCO_2$ vs. AOU, **c** chlorophyll vs. AOU, and **d** chlorophyll vs. nitrate. Colored squares are data between 01-May-2018 and 15-Jun-2018 during the spring bloom. The squares are connected by gray solid lines. A least-squares regression curve is plotted as a solid black line for the time period and the regressions' parameters for the model $y = m \times x + b$ are given in the table at the bottom for the different parameter relations and time periods. Colored diamonds/dashed lines are the same for the time period after the bloom (15-Jun-2018 to 01-Aug-2018). Note that this analysis could not be done in 2017 as the observations were below the productive layer and no nitrate and chlorophyll observations exist.

biomass to the *Phaeocystis* bloom was not detected (Fig. 6c), suggesting that grazing was weak and zooplankton did not act as a quick export vector. Correspondingly, relative sequence abundances of copepods (not shown) increased in June to August, i.e., after the initial bloom. Vertical particle flux throughout the water column was small in May-June 2017 and the flux maxima at 200 m and 1200 m depth were not reached until mid August 2017 (Fig. 6d), i.e., 2 months later than during 2018.

This regenerative system within the stratified layer during 2017 retained the production in the upper ocean for ~4 months (Fig. 6d), likely supporting recycling in the pelagic ecosystem. Peak export arrived at the seafloor in September–November 2017 (Fig. 6d), i.e., ~3 months later than in 2018. Up to 35% of the area was covered by pale-white material (Fig. 6e) characteristic of substantially aged material[46]. *Emiliania huxleyi* accounted for <1% of sequences at 30 m (not shown), and pteropod aragonite typically dissolves before reaching 2000 m depth, suggesting that these factors did not significantly contribute to vertical flux. *Melosira arctica*, which when white in color is aged and nutrient poor[47], contributed to the white material as seen under the

microscope from the lander trap. However, irrespective of what exactly constituted the pale material, its lack of color suggests that labile photosynthetic pigments had been degraded, and the material itself was of highly degraded nature and thus low nutritional quality. This interpretation is supported by the fact that this export event did not elicit a comparable benthic response (one individual megafauna organism present, no change in oxygen consumption; Fig. 6d, e).

## Discussion

In contrast to upper water column oceanography encountered in temperate or tropical open ocean regions, it is the peculiarities of meltwater-induced salinity stratification that are key to the dynamics of the Arctic Ocean. Meltwater can lead to very strong salinity stratification, comparable to the situation in river plumes[48], and it can set in rapidly. Accordingly, our moored observations have recorded the transition from a > 300 m deep mixed layer to a thin mixed layer (<30 m thickness) in less than a month by meltwater effects. Later in the season, the shipboard observations showed stratification to within 5 m of the surface

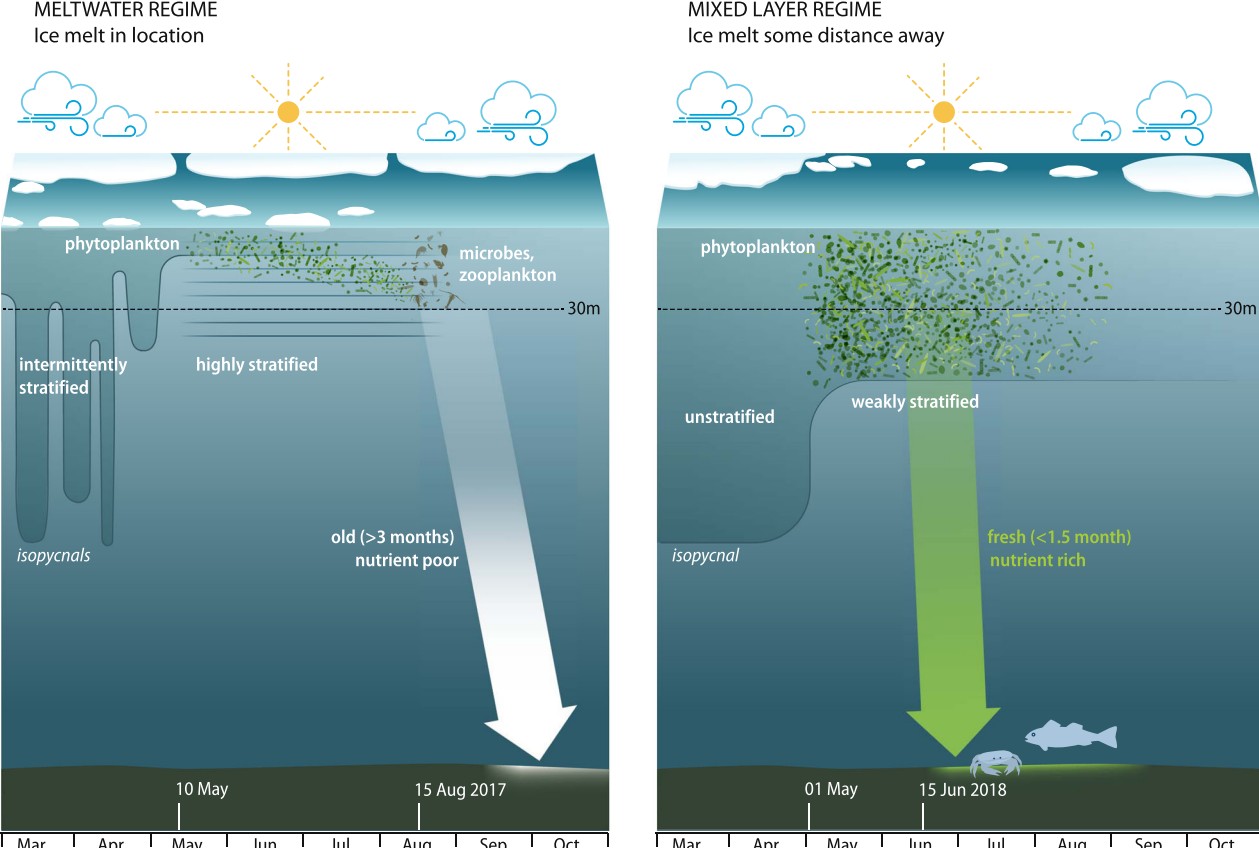

**Fig. 8 Schematic contrasting the Meltwater Regime (left) and the Mixed Layer Regime (right).** The dashed line at 30 m indicates the primary depth of our physical–biogeochemical observations. In addition, we had water column observations in 55 m, 265 m, ADCPs, sediment traps in 200 m and 1200 m, as well as and benthic observations (lander and crawler). We identified two types of surface stratification regimes that resulted in different bloom dynamics and different patterns and rates of carbon export (Table 1). Near-ice, meltwater-stratified waters (MW: meltwater regime observed in 2017 on left) hosted vertically constrained, longer duration blooms, while waters further from the ice edge (ML: mixed layer regime observed in 2018 on right) hosted higher biomass (m$^{-2}$) and shorter, more intense pulses of export carrying significant quantities of algal detritus to the seafloor, which supported higher megafauna densities.

(the mixed layer may have been even thinner than 5 m) with a likely meltwater contribution of up to 1/6 by volume. Temperature-driven stratification can typically not achieve such density differences and can therefore be more easily broken down intermittently by wind induced mixing[49]. The observed surface forcing by winds in 2017–2018 differed somewhat, but had no effect on the evolution of the mixed layer (Fig. S6).

We identified two types of surface stratification regimes that resulted in different bloom dynamics as summarized in Fig. 8 and, notably, different patterns and rates of carbon export (Table 1). Near-ice, meltwater-stratified waters (MW: meltwater regime) hosted vertically constrained, longer duration blooms, while waters further from the ice edge (ML: mixed layer regime) hosted higher biomass (m$^{-2}$) and shorter, more intense pulses of export carrying significant quantities of algal detritus to the seafloor, which supported higher megafauna densities. An additional mechanism that likely contributed to retaining the organic matter in the surface ocean in the MW regime is that sinking rates of aggregates are slowed by strong density gradients[50]. For the same density gradient, this effect is much stronger in a salinity stratified case because of the relatively slower diffusion of salt into aggregates[51]. Our data suggest that it depends on the ice export through Fram Strait whether the hydrographic conditions in the Fram Strait marginal ice zone region follow the MW regime or the ML regime. The exact timing of the onset of the bloom in turn

is determined by when the local heat flux turns positive such that deep winter mixing stops and stratification sets in[52]. According to ERA-Interim, this happened around May 8th with a standard deviation of 8 days between 2010 and 2018.

Our conclusions are based on a temporally highly resolved continuous dataset, which together engender confidence in our description of the ecosystem phenology (i.e., temporal pattern of bloom development) and its drivers. Typical ship-based surveys often miss key transition moments in the system, such as the onset of the spring bloom or the summer/fall bloom in polar regions. Furthermore, the duration over which processes take place can be more fully assessed by autonomous observations that include in-situ estimates of rates and stoichiometries.

Between early March and late August (i.e., before the break-down of stratification in fall), more than twice as much particulate organic carbon (POC) reached the seafloor in the ML regime than in the MW regime (Table 1). We note, however, that the majority of POC export in the MW regime occurred after early September (albeit visibly degraded), so the difference in POC export between early March and late November is small: only one third larger in the ML regime than the MW regime (Table 1) even though the nitrate availability was more than twice as large in the ML than the MW regime (Table 1). The BCP thus was more efficient in the MW regime than in ML regime, meaning that it exported more carbon per unit amount of nitrate exported at the

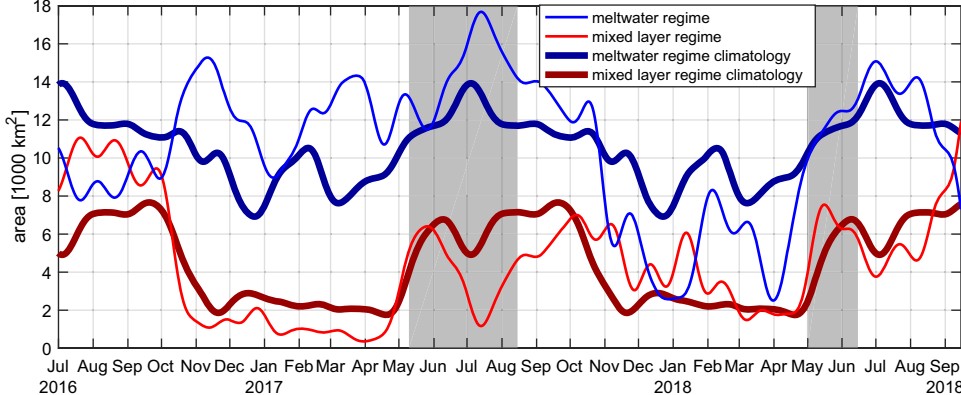

**Fig. 9 Areal coverage of stratification regimes from the numerical ocean model.** Area covered by different stratification regimes within the 18,500 km$^2$ large region 0°E–8°E, 78.5°N–79.5°N (see cyan box in Fig. 1). The meltwater regime's definition in the model is mixed layer depth MLD < 50 m and ΔS > 1 (where ΔS is salinity difference between the surface and 100 m depth). The mixed layer regime's definition is MLD < 50 m and ΔS < 1. The meltwater regime is shown in blue with its climatology in dark blue; it is larger in spring/summer 2017 than typical. The mixed layer regime is shown in red with its climatology in dark red; it is somewhat larger than typical in spring/summer 2018. Unstratified (MLD > 50 m) conditions make up for the rest of the areal coverage.

same time; the nutrients were likely retained in the surface ocean. However, our observations did not take account of the fate of direct sea ice algal production and export early in the year. It was previously observed that during ice melt, sinking velocities of POC and microbial connectivity are higher than in adjacent ice-free waters[53], but this does not hold more generally, in open waters where no ice is available to be melted or over the duration of the productive season[40]. The MW regime we have investigated here is characterized by the meltwater left behind by sea ice that has melted rather than by processes associated with ongoing sea ice melt, i.e., sea ice may no longer be present as it has either fully melted or has been advected out of the area, but the MW regime persists as long as the euphotic zone remains (strongly) salinity stratified.

Our data implies that the meltwater-stratified conditions (MW regime, 2017) favored the longer-term growth constrained within thin layers over multiple months, while the ice unaffected situation (ML regime, 2018) favored a short intense bloom that sank while still rich in chlorophyll, providing fresh labile organic carbon to the benthic ecosystem. In other words, the MW regime might be considered a retention system while the ML regime is more similar to an export system[54–56] and the nature of the physical setting has a measurable impact on pelagic-benthic coupling.

The differences in bloom progression between the two years of observation point to large effects of ice advection (on the scale of 50 km) on the local salinity stratification, ultimately impacting the ecosystem as a whole. We note that the bloom progression in the ML regime (2018) may be representative of future scenarios in the eastern Arctic Ocean subject to projected Atlantification[57]. In those scenarios, salinity stratification is reduced as sea ice may be increasingly absent, even in winter; alternatively, the sea ice and its associated meltwater could be advected out of the area such that the system is no longer directly impacted by ice. Conversely, the MW regime (2017) appears to be representative of blooms in low (but non-zero) ice concentration regions of the parts of the central Arctic Ocean that are covered by sea ice in winter. The sea ice melts prior to complete consumption of the nutrients in the productive layer in spring and produces very strong salinity stratification that cannot be overcome by wind mixing (except possibly for major storms that are not common in the area in summer).

Compared to the ML and MW regimes at mooring HG-IV described above, the variability during those two years observed

55 km further east in purer Atlantic conditions (at mooring F4, Fig. 1, S1–S3) is intermediate with respect to most physical and biological parameters. Likewise, our MW and ML regime years represent the maximum and minimum, respectively, of the typical range of ice area export out of Fram Strait (Fig. 2). Based on results from an eddy-resolving sea ice-ocean model, we estimate that the area in Fram Strait covered by very strong meltwater stratification was almost 4000 km$^2$ larger in spring/summer 2017 than its typical extent (Fig. 9) and it extended further to the east and south. Conversely, in spring/summer 2018 the ML regime covered an area ~2000 km$^2$ larger than typical. Hence, our two years of observations at HG-IV represent end points of the spectrum of the interannual and regional ice variability—and hence meltwater stratification—in this part of the Arctic Ocean up to now.

Climate change will substantially reduce the area covered by sea ice in summer, but in many parts of the Arctic the winter sea ice extent will probably be reduced much less. The seasonal ice zone, in which ice melts in spring and provides strong salinity stratification, could therefore actually become larger over time, increasing the spatial extent of the region experiencing dynamics similar to the MW regime described here. Furthermore, as ice becomes more mobile in the central Arctic, ice area export will likely increase further. However, the observed thinning of sea ice[58] may result in a decrease of the ice volume export[26], meaning that at this stage it is not clear whether areas dominated by ice export such as central Fram Strait will experience an increase (like the MW regime) or decrease (like the ML regime) of the meltwater stratification.

Part of the Atlantic Water present near the surface in eastern Fram Strait subducts below Polar Water in the recirculation in central Fram Strait[20]. This AW then remains isolated from the atmosphere for a long time and partially crosses Denmark Strait to contribute to the formation of North Atlantic Deep Water[59]. Owing to their much larger salinity relative to the surface waters of the MW regime, the surface waters of the ML regime likely participate in this part of the physical carbon pump. This contribution to carbon sequestration needs to be considered in addition to the vertical carbon removal from the atmosphere through the biological carbon pump discussed above.

We note that sudden massive algal falls (of e.g., *Melosira arctica*) likely are rare in the region[7,8,60] and therefore may only

subsidize part of the annual benthic production. While the productivity of larger benthic organisms likely depends primarily on the integrated flux from pelagic blooms throughout the growth season, the quality of less degraded fast-sinking ice algal detritus may also play a role as has been shown in shallow Arctic areas[61]. The benthic community could benefit from an increased frequency of ML regimes, as long as diatoms are winners of the system, and provide intense, nutrient-rich pelagic export events[62]. Where diatoms are replaced by other phytoplankton groups which are not rapidly sinking, the higher productivity may not result in more export flux.

It has been predicted that primary productivity will increase in the future Arctic in most regions, but less so in Fram Strait[63]. Strong stratification in winter in the central Arctic limits nutrient resupply through deep winter mixing and the MW regime inhibits nutrient resupply to the euphotic zone during summer. The specific details of sea ice export, distribution, and melt will critically determine the net impact of global change on Arctic food webs including the benthic ecosystems and need further year-round observations.

## Methods
Table S1 lists the data used in this paper, the instruments that it is based on, the data repositories, and in which figures the data are used.

### Global data sets
*Bathymetry*. Bathymetric data was taken from the International Bathymetric Chart of the Arctic Ocean (IBCAO 30 sec V3)[64] available at https://www.ngdc.noaa.gov/mgg/bathymetry/arctic/grids/version3_0/.

*Sea ice concentration*. We use data derived from the Advanced Microwave Scanning Radiometer sensor AMSR-2 for the years 2013–18 processed in accordance with[65] and downloaded from https://seaice.uni-bremen.de/sea-ice-concentration-amsr-eamsr2/[66]. At each grid point the sum of days during all April/May/June of 2013–2018 when the sea ice concentration at the grid point was >20% was divided by the total number of days with data in those months to obtain the percentage of days with ice concentration >20% (Fig. 1). For separate 7-day periods in April/May/June 2017 and 2018 the mean ice concentration over those 7 days was calculated and the 20% contour of this mean was plotted separately for each of those 7-day periods. For each mooring and each day, the ice concentration at the grid cell closest to the mooring was calculated (Fig. 4a and S1a), and if the ice concentration at the mooring was below 20%, the shortest distance to grid cells where the ice concentration exceeded 20% was calculated (Fig. 4a and S1a). If the ice concentration at the mooring exceeded 20%, the shortest distance to grid cells where the ice concentration was below 20% was calculated and the distance was defined as negative.

*Sea ice velocity and sea ice area export*. Ice area flux estimates in Fig. 2a are calculated using CERSAT (Center for Satellite Exploitation and Research, France) motion estimates together with CERSAT ice concentration information[67]. Fluxes are estimated along a zonal gate positioned at 82°N between 12°W and 20°E and a meridional gate at 20°E between 80.5°N and 82°N (Fig. 1) for the period 1994–2020 (January–May). The ice area flux at the gate is the integral of the product between the meridional and zonal ice drift and ice concentration. For a more detailed description we refer to ref. [68]. Arctic-wide sea ice velocity anomalies (Fig. 2b, c) were computed from the OSI-405-c motion product provided by the Ocean and Sea Ice Satellite Application 635 Facility (OSISAF)[69].

*Satellite chlorophyll*. Surface chlorophyll concentrations measured with the Sentinel 3 A OLCI (Ocean and Land Colour Instrument) were downloaded from https://earth.esa.int/web/sentinel/sentinel-data-access. The 8-day satellite data were averaged for the time series over grid points within boxes of 60 km by 60 km around the moorings.

*Atmospheric reanalysis*. ERA-Interim reanalysis[70] data at the surface on a 0.25° latitude by 0.25° longitude grid at 12 hourly resolution was downloaded from https://apps.ecmwf.int/datasets/data/interim-full-daily/levtype=sfc/. Incoming shortwave radiation (ssr) and outgoing longwave radiation (str), sensible heat flux (sshf), and latent heat flux (slhf) were extracted and averaged to daily values.

### Physical numerical models
*FESOM*. In this study, we used model data from the Finite-Element Sea ice-Ocean Model (FESOM) version 1.4[71]. FESOM is a sea ice-ocean model that solves the hydrostatic primitive equations for the ocean and comprises a finite element sea ice

component. It uses triangular surface meshes for spatial discretization, allowing for a refined mesh in regions of interest, while keeping a coarser mesh elsewhere. In the model configuration used here, a mesh resolution of nominally 1° was applied in the global oceans. The mesh was refined to 25 km north of 40°N, and to 4.5 km in the Nordic Seas and Arctic Ocean. In the wider Fram Strait (20°W-20°E/76°N-82°30′N), the mesh was further refined to 1 km. In this region, the simulation can be considered as eddy-resolving, as the local internal Rossby radius of deformation is about 2–6 km[72,73]. In the vertical, the model used 47 z-levels with a resolution of 10 m in the upper 100 m, and coarser resolution with depth (with a resolution of ~100 m at 800 m depth). For bottom topography, the RTopo-2 data set was used[74]. The model simulation covers the period 2010–2018 and has daily model output. It was forced with atmospheric reanalysis data from Era-Interim[70], and was initialized with model fields from the simulation described in ref. [75]. River runoff (except for Greenland) was taken from the JRA-55 data set[76], and Greenland ice-sheet runoff was taken from ref. [77]. Tides were not taken into account in this simulation. Here we studied the model data of 2016 to 2018 in Fram Strait for comparison with our observations.

*1-dimensional mixed layer depth model*. The PWP[78] 1-dimensional mixed layer model simulates the response of the ocean to surface fluxes. It ignores horizontal gradients and horizontal advection. This allows to judge whether certain surface flux conditions can on their own explain observed conditions. We ran the PWP model (as implemented for Matlab by http://www.po.gso.uri.edu/rafos/research/pwp/) with four different scenarios (Fig. S6: P17-M17, P17-M18, P18-M17, P18-M18) where: P17: An idealized initial profile based on the observed profiles (Fig. 3) representing the conditions in 2017: constant temperature of 2 °C in the vertical, linear salinity gradient from 30.5 at the surface to 35 at 50 m and another linear salinity gradient from 35 at 50 m to 35.1 at 200 m. P18: An idealized initial profile based on the observed profiles (Fig. 3) representing the conditions in 2018: Same as P17 except that the surface to 50 m salinity gradient is from 34.8 to 35. M17: A time series of the the meteorological forcing (10 m wind velocity, heat fluxes, and evaporation minus precipitation) from the ERA-Interim reanalysis (Fig. 4b) at the grid point closest to mooring HG-IV for the period 15-May-2017 to 01-Aug-2017. M18: Same as M17 but for the period 15-May-2018 to 01-Aug-2018. M17 and M18 are provided in Supplementary Data 1.

### Shipboard CTD data
Shipboard CTD casts of a standard dual sensor Seabird 911+ CTD-rosette were occupied in spatial and temporal vicinity to the moored observations (Tab. S2) on three cruises: PS107 in 2017 (https://doi.org/10.1594/PANGAEA.894189), PS114 in 2018 (https://doi.org/10.1594/PANGAEA.898694) of RV *Polarstern*, and JR17005 in 2018 (https://doi.org/10.5285/84988765-5fc2-5bba-e053-6c86abc05d53) of RRS *James Clark Ross*. The data were processed according to standard routine[79]. Additionally, we use underway CTD data from an OceanScience underway CTD collected during PS107 in 2017 (https://doi.org/10.1594/PANGAEA.886146) and processed according to ref. [21].

### Mooring data
The mooring data discussed in this paper is from two mooring clusters in the central and eastern Fram Strait (named "HG-IV" at ~79°N 4°20′E and "F4" at ~79°N 7°E) where moorings were located as close to each other as possible (the horizontal separation was equal to the water depth) in order to enable more measurements than could be fit physically onto a single mooring. Tab. S2/S3 list the deployment and recovery details of the moorings including the exact latitudes/longitudes as well as the individual instruments on the moorings. Note that all data shown in this paper from ~30 m depth and the temperature/salinity/oxygen data from ~55 m is from the HG-IV-S-* and F4-S-* moorings, while all other data is from the HG-IV-FEVI-* and F4-* moorings. The AZFP data is from F5-17 located roughly half way between the two clusters. All sensor based mooring raw data (except for the ASL AZFP data) is available at ref. [80].

It is known that conversion factors for biogeochemical sensors (e.g., chlorophyll fluorescence) change over the seasons, depths, and regions[81,82]. In order to make as few assumptions as possible, we used the following approach: we could have determined the conversion factors from the instance when the ship was there with the CTD-rosette, but these conversion factors might not be appropriate for the majority of the time series. Hence, simply using the manufacturers' calibrations, as we do here, introduces fewer uncertainties. Where we have different estimates of the same parameter, we present them together and demonstrate that they agree qualitatively and also mostly quantitatively (e.g., Fig. 5b). In particular the timing of events is robust.

At some locations, the target variables were not measured the whole time or the measurements failed, hence we present what is available. The vertical location of the instruments (Fig. 4c and S1c) varied substantially (intermittently up to 200 m) as a result of mooring blow downs caused by strong intermittent ocean currents. Time series have not been corrected for this vertical motion, but data are not used during blow downs in order not to bias the time series interpretation by temporal changes introduced by instruments traversing through vertical property gradients.

### Physical sensor measurements
The physical sensors (for pressure, temperature, conductivity, and oxygen) were pre-cruise manufacturer calibrated and processed similar to ref. [83]; the processed data is also available at ref. [80].

*Mixed layer depth (MLD).* Since there are no autonomous vertically profiling measurements available, we can only determine the minimum value of the mixed layer depth. At each hourly time step, the potential density difference ($\Delta\sigma$) between the uppermost (~30 m) temperature/salinity recorder and the underlying temperature/salinity recorders is calculated. The $0.5^{th}$ percentile of each $\Delta\sigma$ time series is added to the $\Delta\sigma$ time series for the different deployments. This fixes slight offsets in the temperature and/or conductivity calibrations which result in too negative or too positive density differences. The minimum estimate of the mixed layer depth at hourly resolution is then determined as the depth of the deepest instrument where $\Delta\sigma < 0.05$ kg m$^{-3}$. If $\Delta\sigma > 0.05$ kg m$^{-3}$ for all depths at a time step, then the minimum mixed layer depth can only be determined as 0 for that time step. Daily values of the MLD were defined as the depth at which three hourly realizations of MLD were shallower within a 24 h time span and at which the remaining 21 MLD realizations were deeper. This biases the daily MLD estimate towards situations where phytoplankton is kept in the surface ocean rather than also being mixed down for some amount of time.

*Stratification estimated between 30 m and 55 m.* Based on the temperature and salinity time series observed at ~30 m and ~55 m, we estimate the buoyancy frequency as $N^2 = \frac{-g}{\rho_0}\frac{\Delta\rho}{\Delta z}$ where $g$ is the acceleration due to gravity, $\Delta\sigma$ is the potential density difference over the vertical distance of $\Delta z = 25$ m, and $\rho_0$ is the average density. The contributions to stratification due to temperature ($N^2_T$) and salinity ($N^2_S$) are estimated as $N^2_T = g * \alpha\frac{\Delta T}{\Delta z}$ and $N^2_S = -g * \beta\frac{\Delta S}{\Delta z}$, respectively, where $\Delta T/\Delta S$ are the temperature/salinity differences and $\alpha/\beta$ are the thermal expansion/haline contraction coefficients estimated from the average temperature/salinity at the two measurement depths.

*Apparent oxygen utilization (AOU).* Oxygen concentration from the microcats was calculated using the pre-cruise manufacturer calibrations. AOU was calculated as the atmospherically equilibrated oxygen concentration (calculated from measured pressure, temperature, and salinity with sw_satO2 from the Seawater toolbox available at http://www.cmar.csiro.au/datacentre/ext_docs/seawater.htm) minus the measured oxygen concentration.

## Light
*Polar night/polar day.* The length of day (hours per 24 h that the sun is above the horizon) was calculated from the sunrise equation as implemented for Matlab by https://de.mathworks.com/matlabcentral/fileexchange/55509-sunrise-sunset.

*Photosynthetically available radiation (PAR).* The WetLabs Eco PAR measured PAR for 5 (2016–2017) or 10 (in 2017–2018) individual measurements 1 s apart from each other before it slept for 1 h before repeating the measurement cycle. These 5 or 10 individual measurements are averaged linearly to obtain hourly values at ~30 m depth (Fig. 5a blue). Values below the detection limit are set to a constant of $10^{-1.32}$ µmol m$^{-2}$ s$^{-1}$. Hourly values are linearly averaged to daily values (Fig. 5a black). The incoming solar shortwave radiation varies as a function of season and latitude as well as cloud cover as represented in the ERA-Interim reanalysis (parameters ssr). Its unit of W m$^{-2}$ is converted to PAR assuming a constant spectral distribution as 1 W m$^{-2}$ = 2.1 µmol m$^{-2}$ s$^{-1}$[84]. In order to compare the PAR measured at a depth of approximately 30 m to the surface values, we approximate a spectrally averaged diffuse attenuation coefficient for PAR in clear water using the values of[85] as $k_d = 0.02$ m$^{-1}$ and apply it to calculate a constant exponential extinction applied to the reanalysis surface values (Fig. 5a yellow). The average PAR available ($PAR_{available}$) to phytoplankton being moved around in the clear water mixed layer of depth *MLD* was calculated as the depth averaged vertical integral of the clear water extinguished PAR at the surface ($PAR_{surf}$ from the shortwave radiation of ERA-Interim): $PAR_{available} = \frac{1}{MLD} * \int_{z=0}^{z=MLD} PAR_{surf} * e^{-k_d z} dz$ (Fig. 5a red).

## Chlorophyll concentration and optical backscattering
*Chlorophyll fluorescence.* The WetLabs ECO Triplet measures fluorescence at a "chlorophyll wavelength" and at a "CDOM wavelength" as well as optical scattering at 700 nm. The conversion from fluorescence to chlorophyll *a* concentration (in µg l$^{-1}$) follows a manufacturer determined conversion determined for a mono-culture of phytoplankton (*Thalassiosira weissflogii*), which typically overestimates the chlorophyll concentration. Hence, we applied the community-established calibration bias of 2 for the WetLabs ECO-series fluorometer to these in situ fluorometric chlorophyll values[81]. This conversion factor may be different in ocean waters of Fram Strait, but it still gives reasonable agreement with independent estimates.

*Optical backscattering.* The EcoTriplet measured 8 individual measurements 1 s apart from each other before it slept for 1 h before repeating the measurement cycle. For the chlorophyll fluorescence, the individual measurements are averaged to hourly values. For the scattering, times when individual 1-second measurements exceed 0.002 m$^{-1}$ sr$^{-1}$ are indicative of strong optical backscattering not due to small particles in the water column, but rather to larger potentially aggregated particles. The times of strong backscattering are marked individually (Fig. 5b red).

## Nutrients
*Nitrate (SUNA sensor).* Prior to deployment (11 and 15 days for sensors deployed at HG-IV and F4, respectively), the reference spectrum of the sensors were updated as per manufacturer specifications. We first let the sensors cool down for 24 h at 0 °C in a temperature controlled laboratory. Next, the reference spectrum update was achieved by measuring Milli-Q water (i.e., no nitrate present). To verify if this update was successful, solutions with three different nitrate concentrations (3, 7, and 14 µmol l$^{-1}$) were then measured, with the output being monitored live (expected to be within ±2 µmol l$^{-1}$ of each concentration). A measuring time of 20 s yields stable results and was thus applied during the deployments with an interval of 6 h. Upon recovery, SUNA data were processed using the SeaBird UCI software package version 1.2.1. Here, temperature and salinity data were used to remove the spectrum of bromide and compensate for temperature dependent absorption using an algorithm developed by ref. [86]. This step yields the spectrum of nitrate only, at a precision of ±0.3 µmol l$^{-1}$. The sensor is characterized by a drift of 0.3 µmol l$^{-1}$ per hour lamp time. Given the deployment settings, a total operational time of about 8 h was accumulated. Therefore, a linear drift correction of 2.4 µmol l$^{-1}$ (365 days)$^{-1}$ was applied. Up to this point, however, accuracy remains at 2 µmol l$^{-1}$ as per manufacturer specifications. Therefore, an offset correction is then applied based on the in situ concentrations observed at the beginning of the deployment as well as with the RAS (see below) where available, with outliers excluded.

*Inorganic nutrients from Remote Access Samplers (RAS).* McLane RAS were programmed to draw two 500 ml samples (1 h apart, starting at noon) approximately every other week. Samples within the RAS were collected in sterile plastic bags and fixed with 700 µl of 50% mercuric chloride solution. Upon recovery, two samples from a given sampling date were combined to yield a volume of 1 l, required for bacterial and phytoplankton genetic analyses (see below), and a 50-ml aliquot destined for the measurement of dissolved inorganic nutrients. Aliquots for nutrient analysis were collected in PE bottles, which were then stored frozen (−20 °C) until analysis on land. Analyses for inorganic nutrients were carried out using a QuAAtro Seal Analytical segmented continuous flow autoanalyser following standard colorimetric techniques. The accuracy of the analysis was evaluated through the measurement of KANSO LTD Japan Certified Reference Materials and corrections were applied accordingly. Finally, we evaluated pressure, temperature, and salinity data from the CTD (SBE37-SMP-ODO) attached to the RAS to determine whether the two samples taken one hour apart on a given date drew water from the same depth and with consistent properties.

## Carbonate system
pCO$_2$ *and pH.* The calibration of SAMI pH and SAMI CO2 sensors was carried out by the manufacturer, approximately 2 months prior to deployment. The calibration certificates specify accuracy and precision of ±0.003/±0.001 pH units and ±3/<1 µatm, respectively. pCO$_2$ sensors measure and yield pCO$_2$ in µatm. For the pH sensor, raw absorption data were converted to pH (total hydrogen ion scale) in combination with temperature and salinity (SBE37-SMP-ODO) using the quality control tool (QC_pH) supplied by the manufacturer.

Upon assessment of pCO$_2$ data, values from the HG-IV mooring in 2017–2018 were deemed to be biased high by approximately 130 µatm (with a step jump at the turn-around of the moorings), therefore a constant value of this magnitude was subtracted from that record. The pH sensors ran out of battery towards the end of the deployments, resulting in interrupted records. At HG-IV, some erratic data before the sensors stopped recording were excluded for the first deployment after 16-Feb-2017 and for the second deployment after 02-Aug-2018. At F4 in 2017–2018 pH values below 8 were excluded.

Carbon takeup is estimated from the change in dissolved inorganic carbon (DIC) between the beginning of the bloom and the time when the minimum pCO$_2$ is reached. In turn, DIC is calculated from pCO$_2$ and alkalinity[87] as well as measured temperature, measured salinity, phosphate concentration (0.5 µmol l$^{-1}$), and silicate concentration (5 µmol l$^{-1}$). Alkalinity is taken from the relationship Alk = 736 + 45.2 * S in ref. [88].

## Microbial communities
*Relative abundances of bacteria and microbial eukaryotes based on 16 S and 18 S rRNA gene sequences.* The methodology followed[89] which we briefly summarize here: the same water samples as for the inorganic nutrients (see above) were used, in which mercuric chloride resulted in fixation of microbes for long-term preservation[90]. After recovery, the ~1 l per sampling event was immediately filtered through 0.22 µm Sterivex filter cartridges (Millipore, Burlington, MA). Filters were immediately frozen at −20 °C until DNA extraction in the home laboratory.

DNA was extracted using the DNeasy PowerWater kit (Qiagen, Germany) according to the manufacturer's instructions and quantified using a Quantus fluorometer (Promega, Madison, WI). Obtained DNA quantities ranged between 0.01 and 11 ng (µl)$^{-1}$. Bacterial 16 S and eukaryotic 18 S rRNA gene fragments were amplified using primers 515F–926 R[91] and 528iF–964iR[92], respectively, according to the 16 S Metagenomic Sequencing Library Preparation protocol (Illumina, San Diego, CA). Amplicon gene libraries were sequenced using Illumina MiSeq instruments at CeBiTec (Bielefeld, Germany; bacteria) or AWI (eukaryotes). After primer removal using cutadapt[93] reads were classified into amplicon sequence variants (ASVs) using DADA2[94].

After singleton removal, we obtained a mean of 60,000 bacterial and 119,000 eukaryotic reads per sample that sufficiently covered community composition[89]. Bacterial and eukaryotic reads were taxonomically classified using the Silva v138 and PR[2] v4.12 databases, respectively.

### Normalized mean volume backscattering (MVBS) as proxy for zooplankton biomass

*MVBS.* RDI Workhorse Longranger ADCPs were deployed, using a four-beam, convex configuration with a beam angle of 20° and frequencies of 76.8 kHz. The number of bins was set to 70 with a bin length of 8 m. The sampling interval was set to 20 pings per ensemble with a ping rate of about 20 pings every 60 min. The instruments were moored at a nominal depth of ~400 m in upward-looking mode and measured horizontal and vertical currents and acoustic backscatter intensity. Instrument heading, pitch and roll and temperature data were also collected. The echo intensities were given in an automatic gain control count scale of 0 to 255. Following[95], they were converted to MVBS. We used beam-averaged data because the four beams together gave a better signal-to-noise ratio than individual beams. First, the noise level of all four ADCP beams was determined from the minimum values of RSSI (Received Signal Strength Indicator) counts obtained in the remotest depth cell, when the sea surface was outside the ADCP range. Sound velocity $c$ and sound absorption coefficient $a$ were considered variable with depth and time and calculated according to the UNESCO formulas from an interpolation in time of 2 CTD profiles collected at the beginning and the end of the deployment. The mean at each depth for the 20 single pings comprising an hourly burst was calculated.

An ASL Acoustic Zooplankton Fish Profiler (AZFP) was moored at F5 (79°N, 5°40′E) with the transducer faces pointing upward in ~150 m depth and operated at four different frequencies (38, 125, 200 and 455 kHz). The sampling interval was set to 30 s with a pulse duration of 0.5 ms (38 kHz) and 0.17 ms (125 and 200 kHz), respectively. Pitch and roll were measured with each ping. Data from the 455 kHz transducer were omitted from this analysis. AZFP data were post-processed and integrated with Echoview software version 11.0.239 (Echoview Software Pty Ltd, Hobart, Australia). Background noise was removed through time-varied thresholds for each transducer, and after removing noise and unwanted signals originating from e.g., other mooring devices that temporally became visible as backscatter in the echograms, the MVBS was integrated and exported for 24-hr bins.

For both ADCP and AZFP data the 50–100 m vertical average was calculated and for the AZFP also the 15–100 m and 30–100 m vertical averages. The median for each mooring deployment of the daily mean values of the vertical MVBS means was calculated. This median was subtracted from the MVBS to obtain the normalized MVBS, which corrects for possible hardware differences between the different deployments[96].

Note that no AZFPs existed at moorings HG-IV and F4 and the ADCPs did not return data shallower than 50 m. Therefore, we use the comparison in Fig. S5 to show that the ADCPs deliver similar qualitative statements compared to the shallower reaching multi-frequency AZFPs. Apart from one exception, there are no large qualitative differences between the averaging in the different depth layers. That means that, most of the time, observations between 50–100 m do not miss a large part of the biomass. The exception occurred in August 2017 when the biomass above 30 m appeared to be much stronger. Likewise, there were no qualitative differences in the progression of the curves between the different frequencies. Therefore, the 50–100 m ADCP appears to be a reasonable proxy of zooplankton and fish biomass and we use it at HG-IV and F4 (Fig. 6c and S3c).

### Particle and POC flux

*Sediment volume flux in water column.* Sediment traps were located 200 m and 1200 m below the sea surface at mooring HG-IV and at 200 m at mooring F4. The collector cup opening times ranged from 9 to 59 days depending on the season (lower resolution in winter). The cups had an interior diameter of 4 cm and the height of the sedimented layer on the bottom was measured from photos of the cups. From these the sediment volume was calculated and it was normalized by the 0.5 m² collection area and the opening duration. We assume that this sediment volume flux is approximately proportional to total particulate matter (i.e., POC and lithogenic matter) flux and use it only to infer qualitative differences and the timing of events.

*POC flux from sediment trap on bottom lander.* Aliquots of the sedimented material collected at 2.5 m above the seafloor in sediment traps on the bottom lander at HG-IV were sieved through 500-µm mesh size to remove larger zooplankton swimmers or benthic organisms, then filtered on pre-combusted Whatman GF/F glass fiber filters, acidified with 0.1 N HCl, and dried at 60 °C. POC concentrations were determined with a CaloErba CN-analyzer. These were then normalized by the split factor as well as to the 0.25 m² collection area and the cup opening duration. As the sediment trap was deployed 2.5 m above the bottom, it also collected resuspended material in addition to settling material.

*POC flux inferred from benthic oxygen consumption.* Benthic carbon mineralization was estimated from sediment oxygen consumption rates. The benthic crawler called TRAMPER[97] was deployed at HG-IV for one year and moved 15 m along the seafloor every 7 days. Upon arriving at the new location, it measured oxygen

concentration profiles through the top 15 cm of the sediment. From the shape of the oxygen profile, the benthic oxygen consumption, i.e., the oxygen flux from the water column into the sediment was calculated. Assuming a Respiratory Quotient of 1.0 (i.e., that $O_2$ consumed via benthic diagenesis is balanced by a corresponding molar production of $CO_2$) oxygen consumption rates were converted to POC fluxes reaching the benthos.

### Seafloor imagery

*Detritus seafloor areal coverage.* An underwater camera (VTLC: Video Time Lapse Camera; AquaPix, USA) fitted to a benthic lander at HG-IV took 5-second video sequences twice a day for the 2017–2018 deployment year. Frame grabs were extracted and divided into 100 boxes referring to equal sized areas of the seafloor. At each time, the number of boxes covered by bare seafloor, by accumulating small white material, and green material was calculated. The white material, though very fine in size, accumulated over a larger area of seafloor than the much more massive green material identified as algal detritus. This is presumably related to hydro-dynamics associated with the microtopography of the seafloor.

*Number of megafauna present.* All eelpout fish (*Lycodes frigidus*) and visible epi-benthic megafauna including shrimps (*Bythocaris* spp), isopods (*Saduria mega-lura*), holothurians (*Kolga hyalina*), and gastropods (*Mohnia* spp.) observed in each frame grab were counted. Additionally, at the very bottom edge of the screen, a purple sea anemone and branches of the sponge *Cladorhiza gelida* were present, but not included in the data.

### Data availability

Data are freely available online at references[98–104] and at the links given in Table S1 or are provided in the supplementary materials file Supplementary_Data_1.xlsx.

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

## Acknowledgements

We thank the FRAM associated technicians, engineers, and administrators as well as the captains, crews, and chief scientists of the cruises who were all instrumental in making these observations possible. We thank T. Kanzow, M. Janout, J. Schaffer, M. Nielsdóttir, and E. Bonk for helpful discussions and contributions to the observational implementation. We thank P. Matrai and an anonymous reviewer for their appreciative reviews. Ship time was provided under grants AWI_PS99_00, AWI_PS100_01, AWI_PS107_05, AWI_PS108_00, AWI_PS114_01, AWI_PS121_01 of RV *Polarstern* and data was also obtained on cruises MSM76 and MSM77 of RV *Maria S. Merian*. The work was supported by the North-German Supercomputing Alliance (HLRN). We thank NOAA/NCEI and the World Data Service for Geophysics for providing the IBCAO bathymetry, the University of Bremen for providing the AMSR2 sea ice concentration, ESA for providing the Sentinel 3 A OLCI chlorophyll concentration, and ECMWF for providing the ERA-I reanalysis. We acknowledge support by the Open Access Publication Funds of Alfred-Wegener-Institut Helmholtz Zentrum für Polar- und Meeresforschung.

## Author contributions

Conceptualization: W.J.v.A., I.S. and A.B. Methodology: W.J.v.A., M.B., C.B., O.B., A.Br., M.H.I., C.K., T.K., N.L., K.M., B.N., E.M.N., A.P., I.S., D.S., T.S., S.T.V., C.W., F.W. and A.B. Formal analysis: W.J.v.A., M.B., C.B., A.Br., B.C., M.H., M.H.I., T.K., K.M., B.N., E.M.N., A.P., I.S., M.S., S.T.V., C.W., F.W. and M.W. Investigation: W.J.v.A., M.B., C.B., J.H., M.H.I., C.K., T.K., N.L., K.M., A.P., I.S., D.S., T.S., S.T.V., F.W. and M.W. Data Curation: W.J.v.A. Writing - Original Draft: W.J.v.A. and A.M.W. Writing—Review & Editing: W.J.v.A., A.M.W., M.B., C.B., O.B., A.Br., B.C., M.H., M.H.I., T.K., K.M., B.N., E.M.N., A.P., I.S., M.S., D.S., T.S., S.T.V., C.W., F.W., M.W. and A.B. Visualization: W.J.v.A., T.K. and C.W. Funding acquisition: T.S. and A.B.

## Funding

Helmholtz Infrastructure Initiative FRAM ("Frontiers in Arctic Marine Monitoring") (A.B.). Ocean Frontier Institute and the Canada First Research Excellence Fund (W.J.v.A., A.M.W.). Open Access funding enabled and organized by Projekt DEAL.

## Competing interests

The authors declare no competing interests.
