## [Peer Review File · Nature Communications]

Sea-ice derived meltwater stratification slows the biological carbon pump: results from continuous observationsReviewers' Comments:

Reviewer #1:

Remarks to the Author:

This manuscript presents a beautiful, multi-parameter data set collected in Fram Strait, a gateway between the Arctic Ocean and the North Atlantic Ocean during two full years. While their time series is longer than that for a few variables, this 2-yr data set covers observations from 30m depth to the ocean bottom. Two bloom and export events (or phases) are described that span the measured range of sea ice flux through Fram Strait, which the authors argue is and will be the major factor in controlling phytoplankton production and subsequent carbon export in this region. Although sea ice flux as well as sea ice extent and concentration in the Fram Strait area have been examined as possible controlling factors of the intensity, timing and duration of the phytoplankton bloom in previous publications (perhaps acknowledge this when the hypothesis is presented, as done earlier in the introduction), none have had access to as highly-resolved temporal data and with as many variables as in the data set described herein. These data will be of great interest to the oceanography, climate, cryology and fisheries research communities, especially when they are publicly available.

There are some model results about stratification, especially to predict the physics of the 0-30m ocean layer where no field data are available from the moorings. While the methods for the parameters are well reported and references provided, few results have error bars, especially when averages are provided. In fairness, Fram Strait, and anywhere where sea ice prevails, is very difficult to sample; so any samples are golden, even if many observational replicates are not available.

I have added many minor comments and questions in the text itself that I will not repeat here. Please address them.

I recommend publication with minor changes.

Patricia Matrai

Reviewer #2:

Remarks to the Author:

The authors provide a multi-disciplinary examination of contrasting spring blooms in Fram Strait from the vantage of a well-instrumented set of moorings. The paper is well written and tells a tidy story that is wonderfully illustrative. I do have some minor editorial comments to provide and suggest some tightening of phrases here and there, but find no major issues that would keep it from publication.

I find myself disliking the approach to the abstract, which avoids describing the actual study and results until more than halfway through. I prefer to leave a broader impacts statement to one concise note at the very end. As applied here, the term "regime shift" in the abstract does not seem to conform to the usage of "regime" used most commonly through the paper, and that is awkward/confusing. The phrase "... hampered by a lack of year round observations across a range of ice conditions" needs further qualifications. I encourage the authors to not inadvertently minimize good work that has been conducted elsewhere, which may not have had as many pieces of equipment but for which one could make a solid case as having conducted a year round sets of ice and BCP-related studies. Because of all of the above points, I would advocate for a substantial re-write of the abstract. Focus on what you learned! I would like to see a quantitative result highlighted... e.g., "We show that, compared to ice-unaffected conditions, sea-ice derived meltwater stratification slowed the BCP by about ZZ%".

Line 97 Here we investigated... For consistency, keep in present tense.

Line 103 add "community" before composition

Line 131 Although it comes out later in the paper, at this first usage you should clarify what you mean by no mixed layer. I take to mean a 0 m thickness, although if you can't resolve one less than 10 m thick then some word-smithing is required. Maybe reword as "no detectable mixed layer".

There are many references to spring, but the MW regime in 2017 extends into July and August, which are both well past spring and into mid/late summer. Some clean-up of the seasonal terminology is warranted throughout the paper. See Lines 135 and 147, amongst others.

Line 267 ML regime (2018). Furthermore, resupply by vertical mixing was very unlikely due to the strong stratification.... You might point to the PWP model here.

Line 309 absence of a mixed layer (<10 m)... even a thin mixed layer should not be conflated with being fully absent.

Line 332 "Our conclusions are based on a temporally highly-resolved continuous dataset, which gives us much greater confidence in our description of the ecosystem phenology (i.e., temporal pattern of bloom development) and its drivers than has ever been possible in the Arctic." ... Yes – you have a wonderful dataset. But statements like "... than has ever been possible in the Arctic" are magnets for readers to seek exceptions - one may find quite a few valid exceptions with some thought! Best to just avoid such statements. You can reword as follows to much better effect, and with tighter writing style: "Our conclusions are based on a temporally highly-resolved and nearly continuous datasets, which together engender confidence in our description of the ecosystem phenology (i.e., temporal pattern of bloom development) and its drivers."

Line 377 very strong salinity stratification that cannot be overcome by wind mixing. ... Possibly too strongly worded. Would not a strong 5-day storm event have a fighting chance to break down some of that stratification?

Line 392: This may be the case for much of the Arctic, but take care not to assume or generalize too much. The Bering Sea ice cover nearly completely collapsed in winter 2017-18 and in this marginal sea it is the winter ice cover extent that exhibits the largest areal change. Only a decade ago one scientist famously predicted that because the Bering would always get cold and dark in the winter there would always be sea ice. That story certainly changed!

I appreciate the extra steps that the authors took to run the PWP model to investigate the differences in MLD. For completeness, I would encourage the inclusion of the 2017 and 2018 forcing function time series so we can see the relative magnitude of the wind stresses, etc.

Line 732 – If you are going to correct the speed of sound based on a CTD profile, maybe a climatological T/S profile would be better in mid-winter than an in situ profile from the deployment or recovery cruise?

Reviewer #1 (Remarks to the Author):

We thank Dr. Matrai for her helpful review and have addressed all the points raised as detailed below.

This manuscript presents a beautiful, multi-parameter data set collected in Fram Strait, a gateway between the Arctic Ocean and the North Atlantic Ocean during two full years. While their time series is longer than that for a few variables, this 2-yr data set covers observations from 30m depth to the ocean bottom. Two bloom and export events (or phases) are described that span the measured range of sea ice flux through Fram Strait, which the authors argue is and will be the major factor in controlling phytoplankton production and subsequent carbon export in this region. Although sea ice flux as well as sea ice extent and

concentration in the Fram Strait area have been examined as possible controlling factors of the intensity, timing and duration of the phytoplankton bloom in previous publications (perhaps acknowledge this when the hypothesis is presented, as done earlier in the introduction), none have had access to as highly-resolved temporal data and with as many variables as in the data set described herein. These data will be of great interest to the oceanography, climate, cryology and fisheries research communities, especially when they are publicly available.

We note that all the data are freely available either in online repositories (Tab. S1) or in the supplementary material (Data S1). In the abstract we have toned down the statements of primacy and at the end of the introduction (L102) we have added the qualifier “Building on work that was partially able to achieve this, we tested the hypothesis that...” in order to highlight that we were not the first to formulate or to attempt to test this hypothesis.

There are some model results about stratification, especially to predict the physics of the 0-30m ocean layer where no field data are available from the moorings. While the methods for the parameters are well reported and references provided, few results have error bars, especially when averages are provided. In fairness, Fram Strait, and anywhere where sea ice prevails, is very difficult to sample; so any samples are golden, even if many observational replicates are not available.

*In fact we have not provided any error bars. One should, however, note that the data of most time-series in our study are hourly (or at least 3-hourly) while we present the data only as 5-days lowpass filtered data. That is, more than 100 (24*5) data points go into most individual numbers shown in the figures and discussed in the text. One could attempt to use this to arrive at the statistical error, but systematic errors would likely be larger, yet unquantifiable.*

In the methods section under “4 Mooring data” at L536-544, we had detailed why they are mostly unquantifiable: “It is known that conversion factors for biogeochemical sensors (e.g. chlorophyll fluorescence) change over the seasons, depths, and regions^{81,82}. In order to make as few assumptions as possible, we used the following approach: We could have determined the conversion factors from the instance when the ship was there with the CTD-rosette, but these conversion factors might not be appropriate for the majority of the time series. Hence, simply using the manufacturers’ calibrations, as we do here, introduces fewer uncertainties. Where we have different estimates of the same parameter, we present them together and demonstrate that they agree qualitatively and also mostly quantitatively (e.g. Fig. 5B). In particular the timing of events is robust.”

While we have tried to be quantitative where possible in the manuscript, the interpretation mostly rests on the timing of events and on order of magnitude estimates and upper/lower bounds. Hence, we do not see how our manuscript would be improved by error bars on one or two occasions (without having them in most other places) and have consequently not added error bars.

I have added many minor comments and questions in the text itself that I will not repeat here. Please address them.

We have copied your comments in the pdf document below and added the line numbers and some context from the original text to make them intelligible when only reading this response document.

We have addressed all your minor comments and have detailed our responses below.

I recommend publication with minor changes.

Patricia Matrai

L68 pls insert comma *Done*.

L92 “a semi-stationary ice edge (here defined as 20% sea ice concentration)” why 20%?
Typically the distinction in behavior is for <10% (open water), 10-80% (some ice), and >80% (ice covered) ice regions, thus whether we choose 20% or another number does not affect the results. We have added a qualifier: “(here defined as 20% sea ice concentration, but not sensitive to the exact definition)”.

L93 “the seasonal migration of the ice edge of 50-100 km” pls add ref
We have added a reference to “Loitering of the retreating sea ice edge in the Arctic Seas” by Steele & Ermold, JGR 2015.

L218 “POC” first use, spell out *Done*.

L245 “pls insert comma” *Done*.

L246 “were” *Done*.

L260 “thin layer (possibly <10 m thickness)”

why 10m?

why not 0-29m above the sensor located at 30m?

We now explain our reasoning in more detail in the text: “Considering the extremely strong stratification to at least 5 m of the surface encountered in the shipboard observations (Fig. 3A/C), we infer that this early primary production resulting in substantial turbidity took place in a thin layer (possibly as thin as 10 m) that limited light in the water below. A euphotic depth of 20 m is achieved by an average chlorophyll concentration of $4 \mu\text{g l}^{-1}$ (as measured by the satellite, Fig. 5B) throughout the euphotic layer (Morel and Berthon 1989). Considering that the satellite possibly underestimates the concentration below the surface and that it averages somewhat in space and time, 20 m is an upper bound on the euphotic depth supporting the speculation that the productive layer may have been as thin as 10 m.”

L264 “very shallow nature of the bloom in highly stratified surface waters and not reaching 30 m.”

true, but the in situ physical measurements don't exist above 30 m. And remote sensing SST is truly surface.

It could be 0-20m.

Certainly above 30m depth

You agree that as written, L264 is correct. Our response to your previous point fleshes out our reasoning.

L265 “The column-integrated nitrate available for production in the highly stratified top layer”
wait how can the 0-30m integration be done w/o surface measurements?

Is this model-derived? If yes, please add "simulated" in front of 'column-integrated'.

Please indicate throughout the text when an estimate or statement is modelled-derived rather than field-data derived.

Except for the outlook on the horizontal extents of the ML and MW regimes (Fig. 8 and accompanying discussion), our paper is entirely based on observational data. However, to avoid confusion, we explain in more detail in L265 how we arrived at the column-integrated nitrate: “There is no reason to suspect that the nitrate concentration above 30 m was higher than at 30 m. Hence, we take the 30-m nitrate concentration and integrate it over the productive layer thickness to arrive at an upper bound on the column-integrated nitrate available for production in the highly stratified top layer; it could have sustained <25 g m⁻² of carbon (Tab. 1)...” In Tab. 1, we have also added the following text to the “Integrated nitrate take-up” row: “No drawdown seen at 30 m; upper bound estimated from nitrate concentration at 30 m.”

L289 “retained the production in the upper ocean for ~4 months”
which Figure? *Added reference to Fig. 6D.*
likely *Added “likely”.*

L292 “*Emiliana huxleyi* accounted for <1% of sequences at 30 m (not shown), and pteropod aragonite typically dissolves before reaching 2000 m depth.”
Are you saying these two factors did not contribute to the vertical flux?
Yes, that is what we meant. We have added this to the text: “..., suggesting that these factors did not significantly contribute to vertical flux.”

L309 “to the absence of a mixed layer (<10 m)”
your data show <30m.
That is still very fast.

True, our moored observations during the restratification phase can only show <30 m. To clarify the point, we added a reference to the shipboard data (Fig. 3): “Accordingly, our moored observations have recorded the transition from a >300 m deep mixed layer to a thin mixed layer (<30 m thickness) in less than a month by meltwater effects. Later in the season, the shipboard observations showed stratification to within 5 m of the surface (the mixed layer may have been even thinner) with a likely meltwater contribution of up to 1/6 by volume. Later in the season, the shipboard observations showed stratification to within 5 m of the surface (the mixed layer may have been even thinner than 5 m) with a likely meltwater contribution of up to 1/6 by volume.”

L321 “that contributes” likely contributed *Done.*

L323 “sinking rates of aggregates are slowed by strong density gradients 49. Waite, A. M. et al. Formation and maintenance of high-nitrate, low pH layers in the eastern Indian Ocean and the role of nitrogen fixation. *Biogeosciences* 10, 5691–5702 (2013).” is this the correct reference?

Waite et al do report about activity within layers in tropical waters, but they did not measure vertical flux in these tropical environment.

I do see the thought extrapolation conveyed by the conceptual model shown in Fig 9; those layers are due to lateral transport at different density depths in the Waite et al ref
We think that the sentence you refer to and the following sentence are both supported by the two references given in the sentences (Waite et al 2013 and Kindler et al 2010). In order to

avoid listing both references at the end of both sentences, we think it is ok to keep the text as originally written: “An additional mechanism that likely contributes to retaining the organic matter in the surface ocean in the MW regime is that sinking rates of aggregates are slowed by strong density gradients ⁴⁹(Waite et al 2013). For the same density gradient, this effect is much stronger in a salinity stratified case because of the relatively slower diffusion of salt into aggregates ⁵⁰(Kindler et al 2010).”

L323 add comma *Done*.

L324 “Our data show that it depends on the ice export through Fram Strait whether the hydrographic conditions in the Fram Strait marginal ice zone region follow the MW regime or the ML regime.”

which fig? where?

Suggest maybe, as coincidence or correlation does not imply causation, right?

Fair! We have thus reworded: “Our data suggest that it depends on...”

L332 “a temporally highly-resolved continuous dataset”

beautiful data set!

A lot of work. *Yep!*

L338 “include in-situ estimates of rates”

true, but not estimated with this data set so far, correct?

No, we have presented the growth rate of chlorophyll a (Fig. S4, L187) based on our time-series measurements.

L348 “nutrients were retained in the surface ocean”

meaning below 30m depth?

Or 0-30m? No data available, unless simulated by model. If the latter, please indicate *Again, this is all observation-based. It is our interpretation that if carbon was exported at a similar rate in both regimes (as observed) even though the nutrient availability was much less in the MW regime, some of the nutrients must have been recycled in the productive layer. Since this is a sentence explaining the concept, we chose not to reword it entirely, but to add the qualifier “likely” to indicate that we have no data to explicitly determine this: “the nutrients were likely retained in the surface ocean”.*

L362 “In other words, the MW regime might be considered a retention system while the ML regime is more similar to an export system and the nature of the physical setting has a measurable impact on pelagic-benthic coupling.”

excellent *Thanks!*

L385 “eddy-resolving sea ice-ocean model”

any biology or biogeochemistry in either of the 2 models used? They appear to be only physical by their description in Methods

Here we have only used physical models. The physical simulations at that resolution have recently been coupled to a biogeochemical model (the implementation is ongoing), but given the scope of the current manuscript and timing we have chosen not to include any of this here. It should enable follow-on studies though. The wording as given in the text (with the

clarification “Physical numerical models” on L474 in our opinion is sufficient information as we are only using information about the stratification in the discussion.

L390 “in this part of the Arctic Ocean”
up to now. *Done.*

L393 “The seasonal marginal ice zone”
do you mean the 'seasonal ice zone', as in the zone where sea ice has been present OR the seasonal variability of the ice edge location per se?
The difference (and its implication) between “marginal ice zone” and “seasonal ice zone” in our opinion is often not clear in general and how it is referred to. Rather than conflating them into “seasonal marginal ice zone”, we agree that it makes more sense to only use either term and now we have done so in the manuscript. Specifically here (L393) and in the abstract (L41), we meant the “seasonal ice zone”.

L395 “increasing the spatial extent of the region experiencing dynamics similar to the MW regime”
this is SIZ for me
We agree, see previous point.

L396-400 this is a rather 'light' section trying to tie together all the pieces linking to sea ice flux in the results section
It is a section of the discussion section of our manuscript, hence we think it is ok if it does not arrive at definitive conclusions, but rather puts things into perspective.

L402-409 This paragraph is not really necessary
The paragraph may not be necessary, but in our opinion it is relevant to not only consider the biological carbon pump, but also the physical carbon pump for the impacts on the global climate. In our opinion it is worthwhile to point out that the region (and regimes) we have focussed on in our manuscript is also important for the physical pump.

L424 “of ice distribution and ice melt”
and sea ice export/flux??
Fair. We have included export and reworded accordingly: “The specific details of sea-ice export, distribution, and melt will...”

L429 “Tab. S1 lists the data used in this paper, the instruments that it is based on, the data repositories, and in which figures the data is used.”
Thank you for providing the data. For the sake of completion, it would be good to add to the excel spreadsheet copies of the various supplemental tables, as lat/lon, sampling depth, and dates are especially needed.
We interpret your comment to mean that we should include Tab. S2 and Tab. S3 also into the Excel document that will be downloadable from the journal’s website (Data S1). We have added them and also Tab. S1.

L474 “Numerical models” physical *Done.*

L662 “Analyses” which ones? Or indicate Tab S1

We meant “Analyses for inorganic nutrients” and have added this to the text.

Tab1 “<<30” ok
ok.

Tab1 “Likely in ~10 m thin layers” no data to support this
Refer to our response to your comment on L260 above. But we have also rephrased to “Possibly in ~10 m thin layers” rather than “Likely”.

Tab1 “chl sensor values” field at 30m or remote sensing?
Clarified as “Based on upper range of chl sensor values at 30 m and 50 m productive layer”.

Tab1 “of the thin productive layer” of the upper productive layer
We are not convinced that “upper productive layer” is better than “thin productive layer”. In fact, neither is required and we shortened to just “productive layer”.

Tab1 “In addition: regenerated primary production from ammonia” what data?
We have no data on ammonia, but wanted to point out that it is something that we have not explicitly considered through our observations. To make that more clear, we have rephrased as “In addition: regenerated primary production from ammonia (not observed)”.

Fig9 great conceptual model *Thanks!*

Reviewer #2 (Remarks to the Author):

We thank reviewer #2 for their helpful review and have addressed all the points raised as detailed below.

The authors provide a multi-disciplinary examination of contrasting spring blooms in Fram Strait from the vantage of a well-instrumented set of moorings. The paper is well written and tells a tidy story that is wonderfully illustrative. I do have some minor editorial comments to provide and suggest some tightening of phrases here and there, but find no major issues that would keep it from publication.

I find myself disliking the approach to the abstract, which avoids describing the actual study and results until more than halfway through. I prefer to leave a broader impacts statement to one concise note at the very end. As applied here, the term “regime shift” in the abstract does not seem to conform to the usage of “regime” used most commonly through the paper, and that is awkward/confusing. The phrase “... hampered by a lack of year round observations across a range of ice conditions” needs further qualifications. I encourage the authors to not inadvertently minimize good work that has been conducted elsewhere, which may not have had as many pieces of equipment but for which one could make a solid case as having conducted a year round sets of ice and BCP-related studies. Because of all of the above points, I would advocate for a substantial re-write of the abstract. Focus on what you learned! I would like to see a quantitative result highlighted... e.g., “We show that, compared

to ice-unaffected conditions, sea-ice derived meltwater stratification slowed the BCP by about ZZ%”.

It was our understanding that for a journal like Nature Communications the motivation of the work starting from the very big picture (broader impacts) should be included in the beginning. Hence the present structure of our abstract. If that was a misunderstanding, we would like the editor to let us know and then we could rewrite the entire structure of the abstract. Currently, however, we have chosen to address the reviewer’s specific suggestions without rewriting the abstract completely and removing relevant information.

To avoid the confusion regarding the word “regime”, we have substituted “undergoes a regime shift” by “changes” in the abstract.

To reduce the impression that previous appropriate work does not exist, we have substituted “Our understanding is hampered by a lack of” by “Full understanding requires” and we have substituted “we present the first such observations” by “we present such observations”. We have included the quantitative result that the BCP is slowed by 4 months and as a further result from our study we include the concept of export vs. retention systems: “meltwater stratification slows the BCP by 4 months, a shift from an export to a retention system, with measurable impacts on benthic communities.” We think the temporal change in the BCP is more relevant than a percent change in the carbon exported, which in fact is not very big despite all the other changes in the system, e.g. changes in the nutrient export.

Furthermore, on L102 we added the qualifier “Building on work that was partially able to achieve this, we tested the hypothesis that...” in order to highlight that we were not the first to formulate or to attempt to test this hypothesis.

Line 97 Here we investigated... For consistency, keep in present tense. *Done.*

Line 103 add “community” before composition *Done.*

Line 131 Although it comes out later in the paper, at this first usage you should clarify what you mean by no mixed layer. I take to mean a 0 m thickness, although if you can’t resolve one less than 10 m thick then some word-smithing is required. Maybe reword as “no detectable mixed layer”.

The CTD and underway CTD data shown in Figure 3A/C contain data to within 5 m of the surface and they do not show a mixed layer. We in fact believe that the mixed layer may have been 0 m, but whether it was 0 m, 2 m, or 5 m is not consequential for the later discussion. Hence, we think that it is still fair to talk of “no mixed layer”. In particular, there is no mixed layer in the sense that most oceanographers would think of when mentioning the term. However, we have added the qualifier “there was no mixed layer (or if it was there it must have been <5 m thick)” to specify what we know.

There are many references to spring, but the MW regime in 2017 extends into July and August, which are both well past spring and into mid/late summer. Some clean-up of the seasonal terminology is warranted throughout the paper. See Lines 135 and 147, amongst others.

It is certainly true that the bloom in the MW regime in 2017 extends into summer. However, in 2018, the bloom terminates on 15-June, which one may consider to be the tail end of spring. Hence we chose to use the following terminology: for referring to the blooms: “spring/summer bloom 2017” and “spring bloom 2018”; and when referring to comparisons between the years irrespective of when the blooms took place: “spring/summer 2017” and

“spring/summer 2018”. We have made these adjustments throughout the manuscript, in particular: L119, L121, L135, L147, L148, L167, L169, L243, L386, L387, Table 1 header, L1197, L1199.

Line 267 ML regime (2018). Furthermore, resupply by vertical mixing was very unlikely due to the strong stratification.... You might point to the PWP model here.

Good point! We have added a reference to Fig. S6 to the text.

Line 309 absence of a mixed layer (<10 m)... even a thin mixed layer should not be conflated with being fully absent.

True, to avoid this conflation we refer to the <5-m thickness in the shipboard data (Fig. 3) while our moored observations during the restratification phase can only show <30 m: “Accordingly, our moored observations have recorded the transition from a >300 m deep mixed layer to a thin mixed layer (<30 m thickness) in less than a month by meltwater effects. Later in the season, the shipboard observations showed a <5-m thin mixed layer with a likely meltwater contribution of up to 1/6 by volume.”

Line 332 “Our conclusions are based on a temporally highly-resolved continuous dataset, which gives us much greater confidence in our description of the ecosystem phenology (i.e., temporal pattern of bloom development) and its drivers than has ever been possible in the Arctic.” ... Yes – you have a wonderful dataset. But statements like “... than has ever been possible in the Arctic” are magnets for readers to seek exceptions - one may find quite a few valid exceptions with some thought! Best to just avoid such statements. You can reword as follows to much better effect, and with tighter writing style: “Our conclusions are based on a temporally highly-resolved and nearly continuous datasets, which together engender confidence in our description of the ecosystem phenology (i.e., temporal pattern of bloom development) and its drivers.”

We fully agree with your point and have used your suggestion.

Line 377 very strong salinity stratification that cannot be overcome by wind mixing. ... Possibly too strongly worded. Would not a strong 5-day storm event have a fighting chance to break down some of that stratification?

Yes, it might. We have added the qualifier to the end of the sentence “(except possibly for major storms that are not common in the area in summer)”.

Line 392: This may be the case for much of the Arctic, but take care not to assume or generalize too much. The Bering Sea ice cover nearly completely collapsed in winter 2017-18 and in this marginal sea it is the winter ice cover extent that exhibits the largest areal change. Only a decade ago one scientist famously predicted that because the Bering would always get cold and dark in the winter there would always be sea ice. That story certainly changed!

Interesting point that we were not aware of. Accordingly, we have qualified this as follows: “but in many parts of the Arctic the winter sea ice extent will probably be reduced much less.”

I appreciate the extra steps that the authors took to run the PWP model to investigate the differences in MLD. For completeness, I would encourage the inclusion of the 2017 and

2018 forcing function time series so we can see the relative magnitude of the wind stresses, etc.

Since the forcing time-series contain many different parameters (and would hence require several subplots), but do not end up accounting for the majority of the difference in the final model profiles, we decided against plotting them. However, we have added the numbers as a separate sheet to the Data S1 supplement and mention this on L509.

Line 732 – If you are going to correct the speed of sound based on a CTD profile, maybe a climatological T/S profile would be better in mid-winter than an in situ profile from the deployment or recovery cruise?

In the stratified periods, the CTD profiles we use would be more appropriate. For winter-time (not the focus of our discussion here) your suggestion would be more appropriate. However, the effects of using either option would miniscule and would not show in the figures at the line width we use or the level of discussion we go for. Hence, we do not apply this suggestion.

Reviewers' Comments:

Reviewer #1:

Remarks to the Author:

Thank you for considering all my comments.

I don't have any additional comments.

I recommend publication as revised.

Reviewer #2:

Remarks to the Author:

I thank the authors for their responsiveness to my review comments and questions. I am satisfied with the revision.